# Anion redox as a means to derive layered manganese oxychalcogenides with exotic intergrowth structures

Shunsuke Sasaki [1,2], Souvik Giri [1], Simon J. Cassidy[1], Sunita Dey[3], Maria Batuk[4], Daphne Vandemeulebroucke [4], Giannantonio Cibin [5], Ronald I. Smith [6], Philip Holdship[7], Clare P. Grey [3], Joke Hadermann [4] & Simon J. Clarke [1] ✉

Topochemistry enables step-by-step conversions of solid-state materials often leading to metastable structures that retain initial structural motifs. Recent advances in this field revealed many examples where relatively bulky anionic constituents were actively involved in redox reactions during (de)intercalation processes. Such reactions are often accompanied by anion-anion bond formation, which heralds possibilities to design novel structure types disparate from known precursors, in a controlled manner. Here we present the multistep conversion of layered oxychalcogenides $Sr_2MnO_2Cu_{1.5}Ch_2$ ($Ch$ = S, Se) into Cu-deintercalated phases where antifluorite type $[Cu_{1.5}Ch_2]^{2.5-}$ slabs collapsed into two-dimensional arrays of chalcogen dimers. The collapse of the chalcogenide layers on deintercalation led to various stacking types of $Sr_2MnO_2Ch_2$ slabs, which formed polychalcogenide structures unattainable by conventional high-temperature syntheses. Anion-redox topochemistry is demonstrated to be of interest not only for electrochemical applications but also as a means to design complex layered architectures.

The rational design of novel compounds is a long-standing goal in solid-state chemistry[1]. For extended non-molecular materials, this is often achieved by (de)intercalation of atomic or molecular species avoiding destructive transformation of the original structure. These topochemical processes are well known in secondary battery electrode materials[2] and for realising functional materials such as superconductors[3,4].

While conventional topochemistry is coupled with cation redox, it was proposed[5] that anionic sp bands are also subject to redox reactions during (de)intercalation processes in highly covalent chalcogenides like $Cu_xCr_2Se_4$[6], $TiS_3$[7], $Li_xFeS_2$[8] and $VS_4$[9]. That concept is gaining ground in battery research aiming to maximise cumulative electrochemical capacity from both cation and anion redox[10]. Anion redox enables extension of chemical space by topochemistry and is

expected to often involve formation or cleavage of anion-anion covalent bonds, for example in the recent reports of $Fe^{3+}[S^{2-}][S_2^{2-}]_{0.5}$[11,12] and $Mg_3V_2S_8$[13], which were respectively derived by Li deintercalation from $Li_2FeS_2$ to form S-S bonds and Mg intercalation into $V^{4+}[S_2^{2-}]_2$ to break them. These structural transformations are drastic despite their reversibility and low reaction temperatures. The recent proof-of-concept work of Cario and co-workers demonstrated that the deintercalation of Cu from LaOCuS led to the conversion of its antifluorite $[Cu_2S_2]^{2-}$ slabs into 2D arrays of $[S_2]^{2-}$ dimers in LaOS[14].

Here, we present multistep deintercalation of Cu from $Sr_2MnO_2Cu_{1.5}Ch_2$ ($Ch$ = S, Se), leading to collapse of antifluorite-type $[Cu_{1.5}Ch_2]^{2.5-}$ slabs into 2D arrays of chalcogenide dimers while retaining the structural integrity of their intergrown $[Sr_2MnO_2]^{2.5+}$ slabs. To

[1]Department of Chemistry, University of Oxford, Inorganic Chemistry Laboratory, South Parks Road, Oxford OX1 3QR, UK. [2]Nantes Université, CNRS, Institut des Matériaux de Nantes Jean Rouxel, IMN, F-44000 Nantes, France. [3]Department of Chemistry, University of Cambridge, Cambridge CB2 1EW, UK. [4]Electron Microscopy for Materials Science (EMAT), University of Antwerp, Groenenborgerlaan 171, B-2020 Antwerp, Belgium. [5]Diamond Light Source, Harwell Science and Innovation Campus, Didcot OX11 0DE, UK. [6]The ISIS Facility, STFC Rutherford Appleton Laboratory, Harwell Campus, Didcot OX11 0QX, UK. [7]Department of Earth Sciences, University of Oxford, Oxford OX1 3AN, UK. ✉e-mail: simon.clarke@chem.ox.ac.uk

achieve this structure transformation, we designed a synthetic pathway focusing on (1) chemoselectivity and (2) the use of reactive synthetic intermediates. Previous use of solvated $I_2$ as oxidant successfully deintercalated ~10% of the Cu from $Sr_2MnO_2Cu_{1.5}S_2$ at 0 °C, enabling control of magnetic exchange pathways between Mn ion moments[15]. Further Cu deintercalation attempted at above 0 °C or using stronger oxidants such as $Br_2$ and $NO_2BF_4$ led to overall decomposition[15], so to circumvent such limitations, oxidants must be chemoselective toward removal of Cu while unreactive to the rest of the oxychalcogenide framework under the reaction conditions. Secondly, synthetic intermediates may circumvent large activation barriers, which are often encountered during topochemical anion redox. For example, the topochemical conversion of LaOCuS to LaOS required a reaction temperature of 200 °C[14] while the conversion of $CeOAg_{0.8}S$ to CeSO, involving $Ce^{3+/4+}$ redox instead of $S^{2-/-}$ redox, proceeded at ambient temperature[16]. Kinetic barriers during anion redox are known to be an adverse factor causing voltage hysteresis in secondary battery systems[17]. Organic syntheses routinely evade such barriered reactions by using energetic intermediates (e.g. via lithiation)[18]. In solid-state chemistry, Kageyama and co-workers proposed the introduction of labile $H^-$ anions in synthetic intermediates enabling access to heavily nitrided[19] or fluorinated $BaTiO_3$[20] at far lower temperatures or pressures than conventional methods.

Inspired by these facts, we designed our synthetic pathway so that the parent phases $Sr_2MnO_2Cu_{1.5}Ch_2$ ($Ch$ = S, Se) are first subject to Li-Cu exchange[21], followed by Li/Cu deintercalation and Cu dissolution using an oxidising agent highly chemoselective toward Cu cation complexation[22]. The final product reveals intergrowth structures consisting of two-dimensional (2D) arrays of dimeric chalcogenide ions separating the transition metal oxide layers.

## Results and discussion

Figure 1a outlines 5-step topochemical conversion of the parent $Sr_2MnO_2Cu_{1.5}S_2$ phase leading to collapse of its copper sulphide layers (see Methods). First reductive Li-Cu exchange was performed following Rutt et al.[21]. The reaction with *n*-butyllithium extruded metallic copper from, and inserted Li into, the parent oxysulfide, giving the mixture $Sr_2MnO_2Li_xS_2$ ($x$ ~ 1.9) + 1.5 Cu (Fig. 1b, see also Supplementary Fig. 1 for the details of scanning electron microscopy (SEM) analyses). Compared to the soft-acidic $Cu^+$ intercalants, the smaller, hard-acidic $Li^+$ cations can be removed more easily from the soft-basic sulphide layers. The Li-containing material was therefore employed as a reactive intermediate to enable metal deintercalation from the sulphide layer. The extruded elemental Cu was then removed using disulfiram (Step 2). The S-S bond of this dithiocarbamate dimer oxidises $Cu^0$ and the reduced anion chelates $Cu^{2+}$ cations[22,23] (Fig. 1a). After Step 2, $Cu^0$ metal was no longer evident in the powder diffraction pattern while the $Sr_2MnO_2Li_xS_2$ phase exhibited cell contraction, implying partial oxidative Li deintercalation by disulfiram (Fig. 1c). A second treatment with *n*-butyllithium (Step 3) extruded about 0.1 equiv. of $Cu^0$ metal by

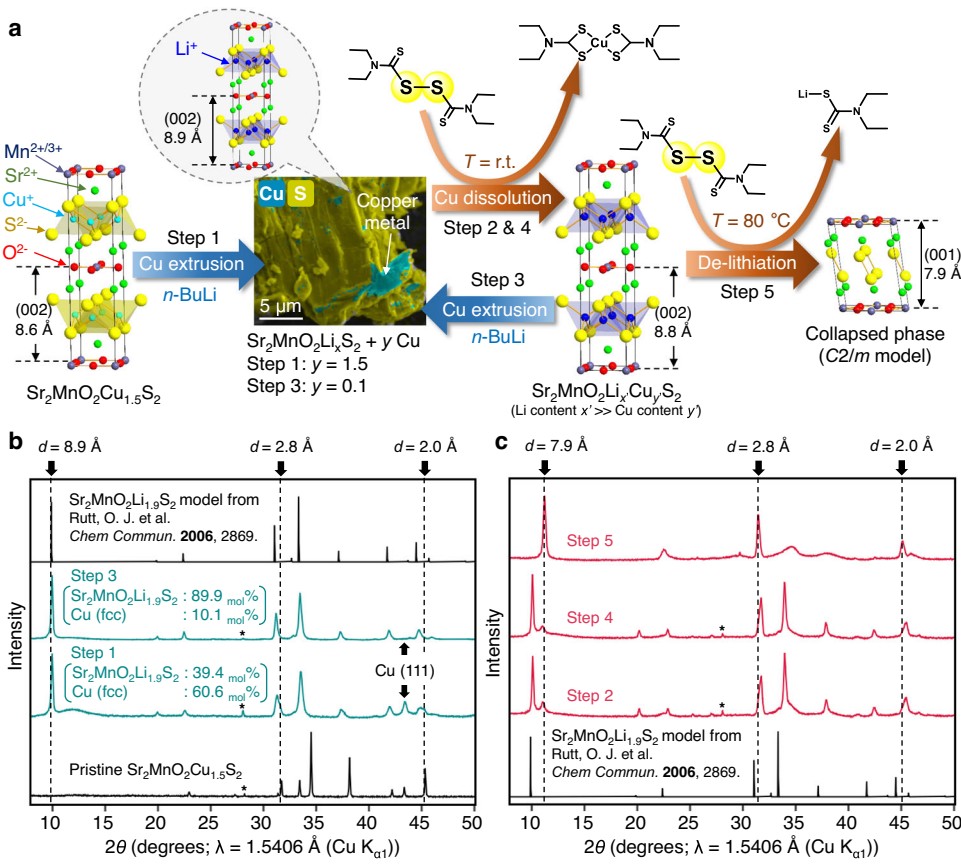

**Fig. 1 | Topochemistry leading to collapse of interlayer spacing between $Sr_2MnO_2S_2$ slabs. a** Synthetic scheme displaying the main reaction at each step and required reagents. Elemental Cu content *y* in the mixture $Sr_2MnO_2Li_xS_2$ + *y* Cu was quantified by Rietveld refinement. The SEM image of the Step1 product is overlaid with the EDX mapping of Cu (cyan) and S (yellow). **b** Laboratory powder XRD patterns of $Sr_2MnO_2Cu_{1.5}S_2$ and its products after first (Step 1) and second (Step 3) treatments with *n*-BuLi at 50 °C. The molar ratio between the oxysulfide and copper was estimated by Rietveld refinements using the $Sr_2MnO_2Li_{1.9}S_2$ model reported by

Rutt et al.[21], whose theoretical pattern is also displayed for comparison. **c** Laboratory powder XRD patterns after first (Step 2) and second (Step 4) treatments with disulfiram as well as after the final treatment at 80 °C (Step 5). Rietveld refinements were performed to account for the sharp peaks in Step 2 and 4 products, fixing the atomic parameters to the $Sr_2MnO_2Li_{1.9}S_2$ structure model. They gave $a = b = 4.00$ Å, $c = 17.62$ Å, which was slightly smaller than $a = b = 4.03$ Å, $c = 17.83$ Å refined for the Step 1 product $Sr_2MnO_2Li_{1.9}S_2$ + 1.5 Cu. * = Si *111* reflection.

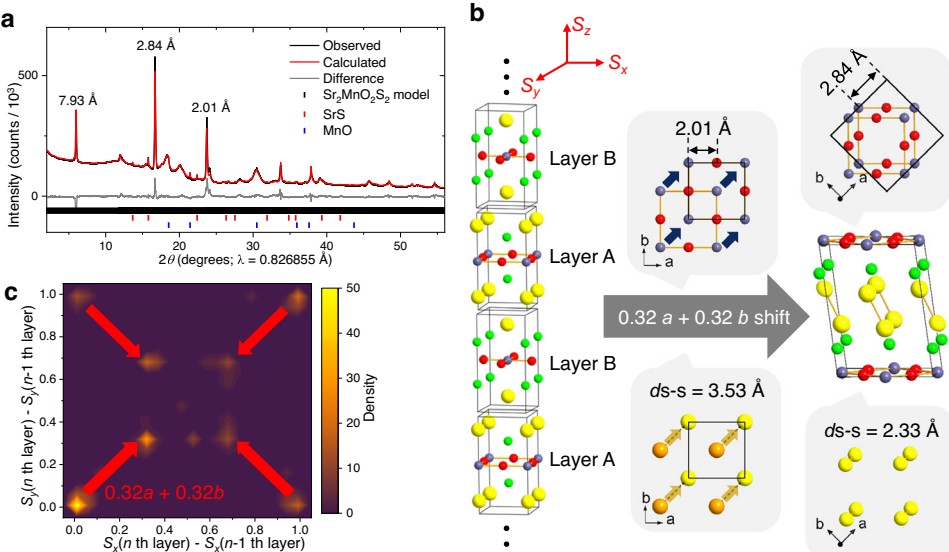

**Fig. 2 | Rietveld refinement and structure modelling of the collapsed phase.**
**a** Rietveld fit to powder PXRD data (I11, Diamond) of the final product of Fig. 1a.
Data (black), fit (red); $R_{wp}$ = 5.74%, $\chi^2$ = 17.1. See Methods section for details. **b** The
structural model used for the refinement. Its unit cell ($a = b$ = 4.030 Å, $c$ = 792.3 Å)
was defined by one hundred $Sr_2MnO_2S_2$ slabs stacked in a body-centred manner.
Sliding of each layer in $xyz$ directions ($S_x$, $S_y$, $S_z$) was parametrised and refined as
fractional values of unit cell parameters. The shift ($S_x$, $S_y$) = (0.32, 0.32) suggested
by the refinement redefined the original $I4/mmm$ cell of the parent phase (middle)
into the new $C2/m$ cell (right, see Supplementary Table 1 for its structure para-
meters). **c** 2D kernel plot representing occurrence of ($S_x$, $S_y$) values refined for
sliding of each $Sr_2MnO_2S_2$ layer (See also Supplementary Fig. 8).

Li re-insertion, suggesting that in Step 2 partial Cu reinsertion to
compensate Li deintercalation competes with Cu dissolution giving
$Sr_2MnO_2Li_xCu_yS_2$ ($x' >> y'$).

Residual Cu metal was then removed using further disulfiram
(Step 4), and a final oxidative deintercalation of Li with disulfiram at an
elevated temperature (80 °C) produced a new product with the pow-
der X-ray diffraction (PXRD) pattern shown in Fig. 1c (Fig. 2a for its
synchrotron PXRD pattern). Sharp Bragg peaks at 2.01 and 2.84 Å
correspond to the $a/2$ and $a/\sqrt{2}$ in-plane dimensions of the $a = 4$ Å
square lattice inherited from $Sr_2MnO_2Cu_{1.5}S_2$, and the sharp peak at
7.92 Å can be assigned to the interlayer spacing between $Sr_2MnO_2$
slabs, which has greatly decreased, so we describe this final product as
a collapsed phase. Other broad peaks in the diffractogram imply the
presence of disorder in the stacking of $Sr_2MnO_2S_2$ slabs. Three-
dimensional (3D) electron diffraction[24] supported severe disorder
along the stacking direction with retention of the structural integrity
within the basal plane (Supplementary Fig. 5). The disorder could not
be removed by thermal annealing but decomposition into multiple
phases began around 300 °C (Supplementary Figs. 6 and 7).

The combination of the lithiation steps and the selective dis-
solution of the extruded elemental Cu was essential to reach the final
collapsed phase. We also treated the parent $Sr_2MnO_2Cu_{1.5}S_2$ phase with
disulfiram without the lithiation step (Supplementary Fig. 2), but the
reaction even at elevated temperature ($T$ = 80 °C) reduced its cell
parameters only to the extent comparable to our previous attempt to
oxidise $Sr_2MnO_2Cu_{1.5}S_2$ using $I_2$ at 0 °C[15], which deintercalated Cu by
about 10%. This result suggests a large activation barrier between the
parent phase and the collapsed phase that must be circumvented by
going via reactive intermediates.

Equally, one must properly design oxidising agents so that they
selectively dissolve elemental Cu without affecting the rest of the host
framework. For example, the reaction of the Step 1 product with $I_2$ in
acetonitrile, a conventional oxidant for chemical deintercalation,
simply restored the parent phase through Li deintercalation and
almost complete Cu reinsertion without any Cu dissolution (Supple-
mentary Fig. 3). The soft-basic, bidentate dithiocarbamate ligands have
a strong preference to form complexes with polyvalent transition

metal cations over monovalent alkali metal cations[25]. In addition, dis-
ulfiram's covalently bonded skeleton made of C, N, S and non-acidic
protons is less likely to destroy the main phase compared to reagents
made of halogens, nitronium or acidic protons. We also tried to dis-
solve $Cu^+$ cations at Step 1 as organolithium cuprates[26] before they are
reduced to elemental $Cu^0$ but these attempts were so far unsuccessful.
Nevertheless, the one-pot removal of Cu would significantly simplify
the synthetic procedure and therefore this will be one of our next
challenges in synthetic methodology.

In contrast to such inaccessibility of the collapsed phase from
$Sr_2MnO_2Cu_{1.5}S_2$, this oxidised phase was stable under air and sponta-
neously formed from $Sr_2MnO_2Li_xS_2$ by aerial deintercalation of Li once
most of the Cu had been removed from the system by the disulfiram.
For example, air exposure of the Step 3 product (i.e. $Sr_2MnO_2Li_xS_2$ + 0.1
Cu) led to emergence of a small PXRD peak at 7.9 Å (Supplementary
Fig. 4), indicating the formation of the collapsed phase as a minor
phase. Similar PXRD patterns were observed also in Step 2 and 4
products (Fig. 1c). These results suggest the presence of the collapsed
phase as the product of aerial surface oxidation.

We modelled the disordered structure by starting from the
$Sr_2MnO_2Cu_{1.5}S_2$ structure, removing the $Cu^+$ ions, and reducing the
interlayer spacing to 7.92 Å (Fig. 2b). A unit cell containing one
$Sr_2MnO_2S_2$ slab was then used to generate a one-hundred-layer
supercell, with the body centring of the parent phase mimicked by
offsetting alternate layers by ½ in $x$ and ½ in $y$. Atomic positions within
each slab were fixed to respect the structural parameters of the ori-
ginal $Sr_2MnO_2Cu_{1.5}S_2$ phase. In the refinement, each $Sr_2MnO_2S_2$ slab
was given individual freedom to shift in $x$ and $y$ relative to the next slab
as well as in the $z$ direction, allowing layers to get closer together or
further apart along the $c$ axis. The refinement gave satisfactory fits to
both X-ray (Fig. 2a) and neutron powder data (Supplementary Figs. 9
and 10) and suggested a high probability of $Sr_2MnO_2S_2$ slabs sliding
relative to adjacent layers by $0.32(3)a + 0.32(3)b$, as well as those
remaining at the original relative positions (Fig. 2c). There was a strong
correlation between a lack of shift between slabs in the $ab$ planes and
larger layer separations (Supplementary Fig. 8) consistent with
remaining parent-like $Sr_2MnO_2Cu_{1.5}S_2$ layers. The occurrence of shifts

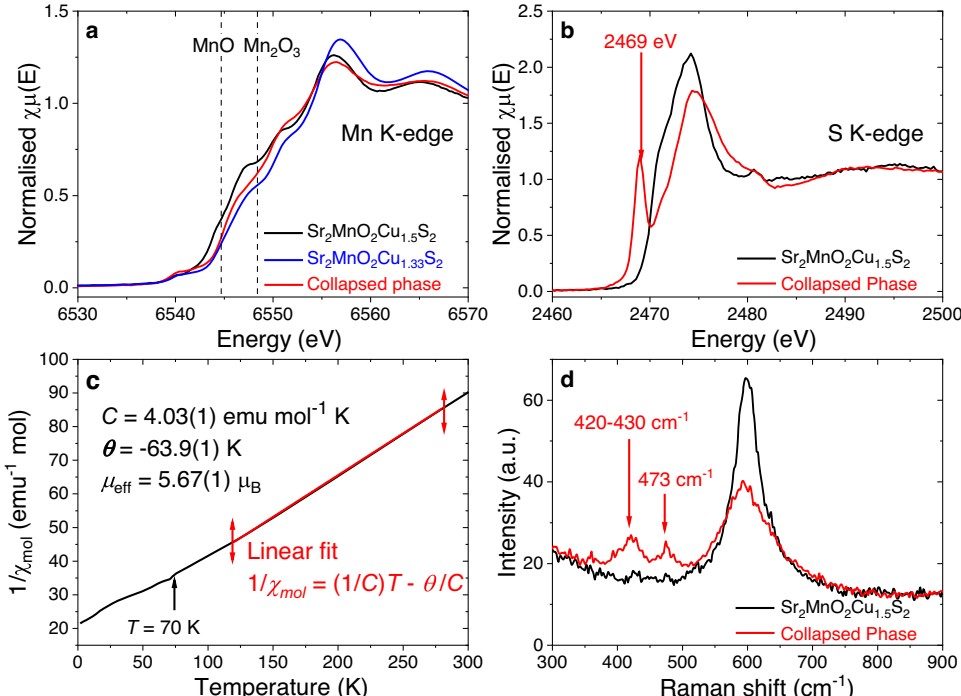

**Fig. 3 | Experimental evidence of sulphur oxidation and disulphide bond formation. a, b** XANES spectra at the Mn K-edge (**a**) and S K-edge (**b**) measured for pristine $Sr_2MnO_2Cu_{1.5}S_2$ (black), $Sr_2MnO_2Cu_{1.33}S_2$ (blue, reproduced from the data published in ref. [15]), and the collapsed phase (red). Dotted lines represent the absorption edges of reference standards MnO and $Mn_2O_3$, which correspond to the lowest-energy maxima on the $d(\chi\mu(E))/dE$ plot (See ref. [30] for their spectra). **c** Inverse molar susceptibility $1/\chi_{mol}$ of the collapsed phase plotted against temperature ($2\,K \leq T \leq 300\,K$). Magnetisation (emu) was measured at the applied field

$H = 100$ Oe after zero field cooling (ZFC) and converted into $1/\chi_{mol}$ (emu$^{-1}$ mol Oe) supposing the tentative formula $Sr_2MnO_2S_2$ (molar mass = 326.31). The linear region ($120\,K \leq T \leq 280\,K$) of the obtained $1/\chi_{mol} - T$ plot was fitted by Curie–Weiss law $1/\chi_{mol} = (T/C) - (\theta/C)$, where $C$ and $\theta$ represented the Curie and Weiss constants, respectively. See Supplementary Fig. 17 for other magnetic data. **d** Raman spectra of $Sr_2MnO_2Cu_{1.5}S_2$ (black) and the collapsed phase (red). The measurement was performed under air with excitation wavelength $\lambda_{ex} = 532$ nm. See Supplementary Fig. 19 for the spectra measured at $\lambda_{ex} = 785$ nm.

of 0.32 along the $a + b$ direction gives a new structure defined by the space group $C2/m$ with a $\sqrt{2}a \times \sqrt{2}b$ cell expansion (Fig. 2b). This $C2/m$ model correlates with smaller interlayer separations and features S-S distances shortened down to 2.33 Å.

Instead of allowing each $Sr_2MnO_2S_2$ slab to slide freely, we also attempted two other approaches that constructed large supercells by piling up the pre-defined collapsed disulphide layers and $Sr_2MnO_2Cu_{1.5}S_2$-like layers (See Supplementary Figs. 11 to 16 for further discussion). Although these approaches did not give better fitting than Fig. 2a, they gave the qualitative idea that both types of layers coexisted in significant amounts and in various stacking sequences. This point is revisited later in the article with our microscopy and spectroscopic analyses.

The obtained $C2/m$ model exhibited S-S distances significantly shorter than those of the original $Sr_2MnO_2Cu_{1.5}S_2$ phase. When atomic coordinates within a $Sr_2MnO_2S_2$ slab were refined during stacking fault analyses (see Supplementary Table 2 for the refined parameters), S-S distances shortened further down to 2.18-2.21 Å. These values are comparable to the 2.13 Å and 2.16 Å in $BaS_2$ and $FeS_2$ respectively[27,28], suggesting S-S dimers are present in the collapsed phase derived from sulphide oxidation. This was supported by spectroscopic evidence. Figure 3a, b compare X-ray Absorption Near Edge Structure (XANES) spectra of $Sr_2MnO_2Cu_{1.5}S_2$ and its collapsed derivative. The Mn K-edge absorption in the collapsed phase increased in energy by only 0.2 eV (Edges in MnO and $Mn_2O_3$ differ by 4 eV)[29], and was even smaller than the 0.6 eV shift on partial Cu deintercalation to $Sr_2MnO_2Cu_{1.33}S_2$[15]. A Curie-Weiss fit to the magnetic susceptibility (Fig. 3c) of the collapsed phase produced an effective moment $\mu_{eff}$ of 5.67(1) $\mu_B$, similar to $Sr_2MnO_2Cu_{1.5}S_2$[30], consistent with retention of a mean Mn oxidation state of +2.5.

In contrast, sulphide anions showed clear sign of oxidation. The S K-edge XANES of the collapsed phase (Fig. 3b) featured an increase in energy of its main absorption peak by ca. 0.6 eV and emergence of a pre-edge peak at 2469 eV, similar to that in $BaS_2$ (but not present in BaS) (Supplementary Fig. 18) which is diagnostic for oxidised $[S_2]^{2-}$ species, corresponding to a transition to antibonding σ* levels[31–33]. Two Raman bands occur in the region 390–510 cm$^{-1}$ (Fig. 3d), typical of sulphide dianions $[S_x]^{2-}$ stretching modes[27]. Steudel's empirical relationship between S-S bond length $d_{SS}$ (Å) and wavenumber $\nu_{SS}$ (cm$^{-1}$)[33]: $d_{SS} = 2.57 - 9.47 \times 10^{-4} \cdot \nu_{SS}$ gave $d_{SS} = 2.12$ and 2.16-2.17 Å for $\nu_{SS} = 473$ and 420–430 cm$^{-1}$, respectively. The two bands implied the presence of two slightly different species of sulphide dimers in the sample.

Despite the evidence for full metal deintercalation from sulphide layers, bulk chemical analysis by Inductively Coupled Plasma Mass Spectrometry (ICP-MS) (Supplementary Table 3) revealed residual Cu and Li after the 5th step (Fig. 1a). Mole ratios of Sr/Mn/Cu/Li = 2.00/1.07/0.35/0.13 (Sample 1) and 2.00/1.07/0.24/0.23 (Sample 2) revealed 0.5 equiv. of residual Cu and Li per Mn ion. SEM analyses showed heterogeneous distributions in 10-100 μm sized crystals (Supplementary Fig. 20), with higher residual Cu content in the middle of the crystallites (Fig. 4a–c).

Electrochemical intercalation of Li into Sample 1 ($Sr_2MnO_2Cu_{0.35}Li_{0.13}S_2$) displayed a 2.0 V (vs. Li$^+$/Li) plateau corresponding to insertion of ~0.5 equiv. of Li, followed by a long sloping process before reaching a capacity equivalent to about 1.75 mol of Li per formula unit at 0.5 V (Fig. 4d). This contrasts with the electrochemical behaviour of the parent phase $Sr_2MnO_2Cu_{1.5}S_2$, where the long sloping plateau corresponding to Li-Cu exchange was observed at a much lower potential (-1.3 V)[34]. This implies that a different redox process is occurring during Li insertion into the new collapsed phase prepared in

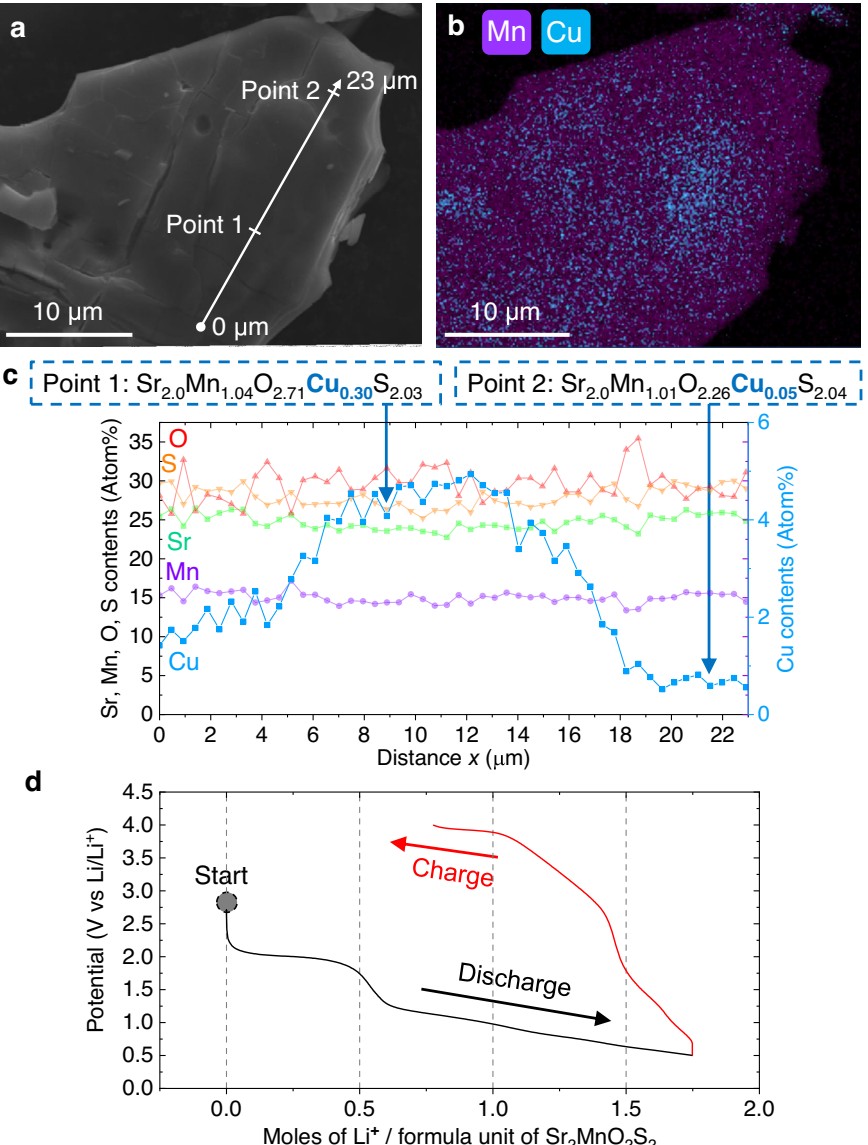

**Fig. 4 | Characterisation of residual Cu and Li in the collapsed phase.**
**a** Secondary electron (SE) image of Sample 2 (bulk composition estimated by ICP-MS: $Sr_{2.00(2)}Mn_{1.07(2)}O_2Cu_{0.24(0)}Li_{0.23(1)}S_2$) and **b** its EDX mapping overlaying signals from Mn $K_{\alpha1}$ and Cu $K_{\alpha1}$ lines. EDX mappings of other elements are found in Supplementary Fig. 21. **c** Sr, Mn, O, Cu and S contents plotted over the 23 μm line section shown in Fig. 4a. To obtain better precision than these line-scan data, single-

point EDX spectra were acquired separately at the positions corresponding to $x = 8.9$ (Point 1) and 21.5 μm (Point 2) of the line section. **d** Voltage vs specific capacity plot for a half cell constructed with Sample 1 (bulk composition estimated by ICP-MS: $Sr_{2.00(2)}Mn_{1.07(2)}O_2Cu_{0.35(0)}Li_{0.13(0)}S_2$) as the positive electrode and Li metal as the anode, cycled at a C/10 rate.

this work. The insertion of 1.75 mol of Li during insertion is consistent with the composition estimated by ICP-MS for Sample 1 ($Sr_2MnO_2Cu_{0.35}Li_{0.13}S_2$), assuming that the Li content after its electrochemical lithiation is the same as for direct lithiation of $Sr_2MnO_2Cu_{1.5}S_2$[21]: $Sr_2MnO_2Cu_{0.35}Li_{0.13}S_2 + 1.75 \; Li^0 \rightarrow Sr_2MnO_2Li_{1.88}S_2 + 0.35 \; Cu^0$. The similar recovery of the original $I4/mmm$ structure type was confirmed when the collapsed oxysulfides were treated with 1.0 equiv. of n-butyllithium solution (ca. 0.1 M in n-hexane) at ambient temperature (Supplementary Fig. 22).

The subsequent charging step in Fig. 4d exhibited a large voltage hysteresis and a capacity of only 60% of that obtained during discharge. Such a path hysteresis is common in electrochemistry involving anion redox, with long-standing debates about its mechanisms[12,35–38]. One explanation ascribed energy inefficiency during anionic oxidation to sluggish structural distortions[17], which sometimes took days or weeks to reach equilibrium[39]. Similarly, the

voltage hysteresis in the current system can be ascribed to the large structural rearrangement associated with the formation of the collapsed phase and its likely high associated activation energy. The presence of two electrochemically active centers, i.e. $Mn^{2+/3+}$ and $S^{2-/-}$ redox, potentially enables charge and discharge processes to take different pathways of structure transformation, leading to a voltage hysteresis[38,40]. The presence of residual Cu may also play a role in this hysteresis, as seen previously in the Cu-system[41]. In any case, such kinetic barriers may also explain why the chemical deintercalation at Step 6 (Fig. 1a) could not remove all the Cu/Li cations from the final product. Further in-depth analyses of its reaction dynamics are currently ongoing, employing in- or ex-situ spectroscopic and diffraction techniques.

High-Angle Annular Dark-Field Scanning Transmission Electron Microscopy (HAADF-STEM) images provided further insights about how residual Cu and Li were present within parts of the collapsed

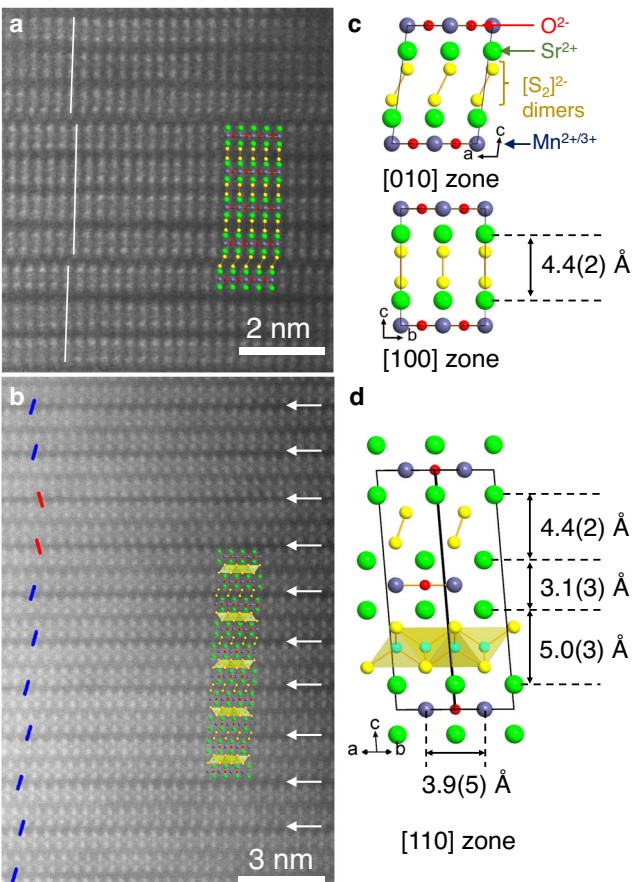

**Fig. 5 | High-angle annular dark-field (HAADF) scanning transmission electron microscopy (STEM) images of Sample 1 ($Sr_2MnO_2Cu_{0.35}Li_{0.13}S_2$).** **a** Close-up view displaying intergrowth of [100] and [010] zones of the collapsed phase (See Supplementary Fig. 23 for the whole image). The structure of the $C2/m$ model (Fig. 2b) was overlaid for comparison. **b** Close-up view of the domain where the collapsed-type layers intergrow with the parent-type layers. White arrows highlight darker and thinner strips that represent the metal-free layers of disulphide ions. Between the collapsed-type layers, $Sr_2MnO_2$ slabs shift either left or right (highlighted by red or blue lines, respectively). The overlaid image depicts the tentative structure model of the intergrowth domain, where structural parameters within the respective intergrowing slabs are fixed to those of the $C2/m$ model and $Sr_2MnO_2Cu_{1.5}S_2$[30]. See Supplementary Fig. 24 for details. **c, d** Structural parameters obtained from the convoluted intensity profile (Supplementary Fig. 25) of Fig. 5a, b. The distances between planes of $Sr^{2+}$ ions are shown.

phase samples. Figure 5a shows the stacking of $Sr_2MnO_2$ slabs, whose interlayer spacing (Fig. 5c) was comparable with the $C2/m$ model of the collapsed phase (Fig. 2b). The image directly showed relative sliding of the $Sr_2MnO_2$ slabs along a diagonal direction (i.e. the 0.32 shift along the $a + b$ direction resulting in monoclinic symmetry). Besides the collapsed phase, we also observed domains containing the original $I4/mmm$ structure with Cu or Li in its antifluorite-type metal sulphide layers and its intergrowth with the collapsed-type slabs (Supplementary Fig. 24). Figure 5b highlights the intergrowth domain; it features periodic dark strips separating pairs of $Sr_2MnO_2$ slabs with a spacing between pairs comparable to that in the collapsed $C2/m$ structure. Each pair of $Sr_2MnO_2$ slabs resembled fragments of the parent-type structure with structural parameters matching well with those of $Sr_2MnO_2Cu_{1.5}S_2$. This indicates the prospect of the 1:1 intergrowth phase $[Sr_2MnO_2S_2][Sr_2MnO_2(Cu,Li)_{1.5-x}S_2]$ depicted in Fig. 5d. Since the intergrowth phase was observed next to the parent phase, the 1:1 periodic structure can be interpreted as the opposite of Daumas-Hérold type ordered staging proposed for intercalation processes[42,43].

To summarise, residual Cu/Li intercalants were present in extremely diverse conditions, including not only regions of the parent-type $I4/mmm$ structure but also its intergrowth with the collapsed type slabs.

The presence of both parent phase and intergrowth layers alongside the collapsed layers encouraged us to attempt Rietveld refinements explicitly taking the presence of these residual Cu and Li cations into account. A custom-built code generated ~20000 unique 1500-layer supercell models containing different combinations of those three structures (see Supplementary Note 1 for details). Each of these base supercell models were refined against the PXRD data, in a procedure similar to that of Bette et al.[44] However, this did not produce convincing fits. Thus there is no single set of parameters to define the homogeneous mixing between the three phases at any given point. Instead, the data are more consistent with each crystallite containing some regions which are highly mixed and some which are sparsely mixed. Such heterogeneous distribution of intercalated layers was reflected also in the Energy Dispersive X-ray (EDX) mapping in Fig. 4a–c. Bulk probes like PXRD are limited in their capacity to analyze stacking faults in such spatially heterogeneous samples, and they must be complemented by local probes.

To investigate the local environments of residual Li ions, $^7Li$ Nuclear Magnetic Resonance (NMR) spectroscopy was performed on the collapsed oxysulfide (Sample 1: $Sr_2MnO_2Cu_{0.35}Li_{0.13}S_2$) and its lithiated derivatives. Before lithiation, the collapsed oxysulfide featured multiple peaks (Fig. 6a) consistent with the complexity of the diffraction results, while both electrochemically and chemically lithiated samples exhibited much simpler spectra with lower frequency resonances (Fig. 6b, c). Large hyperfine shifts result from transfer of unpaired electron spin density from paramagnetic $Mn^{2+/3+}$ ions to lithium s orbitals[45,46]. Our analysis of $^{6,7}Li$ spectra of $Sr_2MnO_2Li_xS_2$ and structural homologues estimated that each $Mn^{2+}$-S-Li interaction resulted in a shift of 101 ppm[34] while a $Mn^{3+}$-S-Li interaction gave a larger shift of 165 ppm due to greater $Mn^{3+}$-S covalency[41]. Since each Li is surrounded by four neighbouring Mn in the parent-type structure (Fig. 6d), and hyperfine shifts are additive, four $Mn^{2+}$-S-Li interactions should result in a hyperfine shift of ~404 ppm, increasing to 468 or 532 ppm when one or two of the Mn ions are oxidised to the 3+ state. These predicted values can be used to assign and rationalise the spectra of the lithiated phases and the lowest frequency resonances seen in Fig. 6a for the collapsed phase.

Higher frequency resonances in the spectrum of the collapsed phase (Fig. 6a), could be rationalised by taking into account the Mn-$[S_2]^{2-}$ coordination in the intergrowth structures (Fig. 6d). Weaker, longer Mn-S bonds are expected for Mn-$[S_2]^{2-}$ compared with Mn-$S^{2-}$ bonds, due to the lower charge of the $[S_2]^{2-}$ ion and its higher energy π* antibonding orbitals that overlap directly with the Mn 3d orbitals. So a Mn ion asymmetrically bound to one $S^{2-}$ and one $[S_2]^{2-}$ anion, will form a stronger covalent bond to the single $S^{2-}$ ion than when it is bound symmetrically to two $S^{2-}$ ions and we deduce that an asymmetric Li-$S^{2-}$-$Mn^{2+}$-$[S_2]^{2-}$ bond pathway gives rise to a larger shift of approximately 165 ppm, compared with 101 ppm for the Li-$S^{2-}$-$Mn^{2+}$-$S^{2-}$ pathway. Figure 6d shows how the presence of intergrowth structures (Fig. 5d) accounts for the multiple higher frequency resonances in Fig. 6a. A 1:1 intergrowth with more Li-$S^{2-}$- $Mn^{2+}$-$[S_2]^{2-}$ bond pathways result in higher shifts than 1:2 intergrowth structures, and $Mn^{3+}$ gives higher shifts than $Mn^{2+}$. The larger number and wide variety of Li environments in the complex faulted collapsed oxysulfide, are all converted into those of $Sr_2MnO_2Li_2S_2$ structure upon lithiation with reduction of both Mn and $[S_2]^{2-}$.

The multiple high frequency peaks above around 600 ppm disappear after chemical or electrochemical lithiation (Fig. 6b, c). Their NMR shift could fully be explained by the parent type structure with different $Mn^{2+/3+}$ ratios showing hyperfine interaction via $Mn^{2+/3+}$-S-Li interactions. This suggested that most of the collapsed type stacks were removed upon lithiation and its product no longer suffered from the severe stacking disorder.

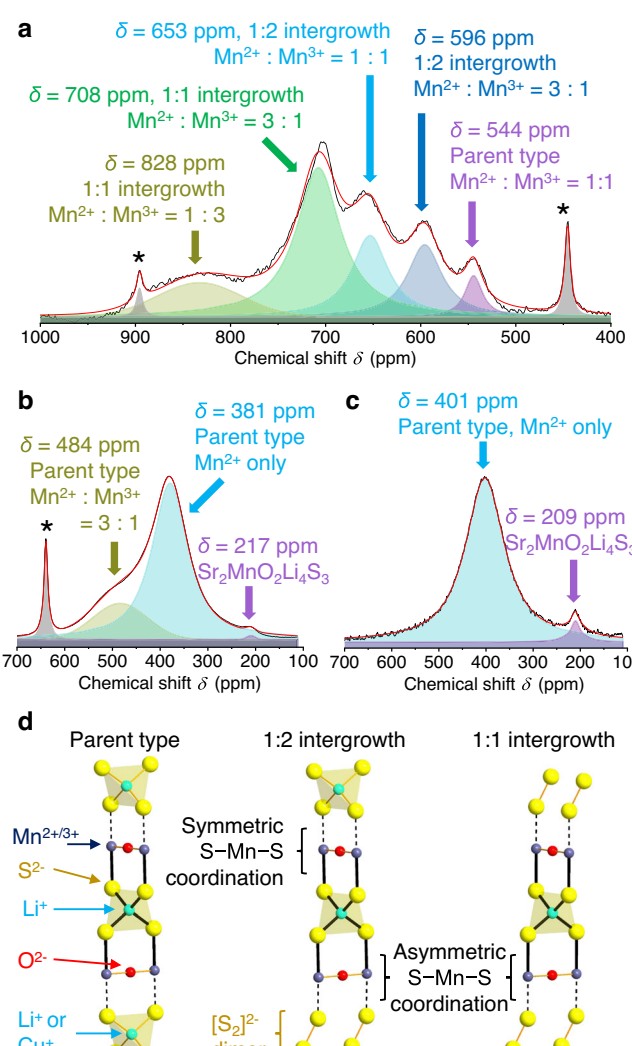

**Fig. 6 | $^{7}$Li NMR spectra before and after lithiation. a** $^{7}$Li NMR spectrum recorded for the collapsed oxysulfide phase (Sample 1) and **b** after its electrochemical lithiation (i.e. after discharge down to 0.5 V (vs. Li/Li$^{+}$) corresponding to the intercalation of 1.75 equivalents of Li; see Fig. 4d) and **c** after chemical lithiation (i.e. after addition of 1.5 equivalents of *n*-BuLi; see Supplementary Fig. 22). Each spectrum was deconvolved (black line: observed, red line: fit) into Gaussian or Lorentzian peaks and their hyperfine shifts were interpreted in terms of the Li local structure as well as the Mn$^{2+}$/Mn$^{3+}$ ratio. Sidebands of the peak at 0 ppm (due to diamagnetic impurities) are labelled with a asterisk (*). See Supplementary Fig. 26 for the whole spectra. **d** Three possible local environments around Li. Black thick lines highlight connectivity between Li and four nearby Mn$^{2+/3+}$ cations. Mn$^{2+/3+}$ cations are either sandwiched symmetrically by S$^{2-}$ anions at apexes of tetrahedra, or asymmetrically by S$^{2-}$ apexes and S$_2$ dimers. In the latter case, charge transfer from S$_2$ dimers to Mn leads to stronger Mn-S-$^{7}$Li interactions.

The chemistry described in Fig. 1a was applied to the oxyselenide homologue. (See Supplementary Figs. 27 and 28 and the Methods section for details). Figure 7a shows the powder pattern which was modelled in a similar way to that of the collapsed oxysulfide (Fig. 2) with collapse of the interlayer spacing between Sr$_2$MnO$_2$Se$_2$ slabs from 8.97 Å$^{47}$ in Sr$_2$MnO$_2$Cu$_{1.5}$Se$_2$ to 8.34 Å and the presence of stacking disorder which was modelled as described above for the oxysulfide (Fig. 7b). The Se-Se distance (Fig. 7c) of 2.43 Å is in the typical range for polyselenides ($d_{Se-Se}$ = 2.35–2.55 Å)$^{48}$, and oxidation of Se$^{2-}$ anions into anionic [Se$_2$]$^{2-}$ dimers was confirmed by the pre-edge feature at 12657 eV and shift of the main peak by ca. 3.4 eV in the Se K-edge XANES (Fig. 7e)$^{49}$. The Mn K-edge XANES hardly shifted compared with Sr$_2$MnO$_2$Cu$_{1.5}$Se$_2$ (Fig. 7d) and the effective moment $\mu_{eff}$ = 5.32(1) $\mu_B$

given by magnetometry (Supplementary Fig. 30) remained in between the spin-only values for Mn$^{2+}$ and Mn$^{3+}$. A Raman peak at 282 cm$^{-1}$ (Fig. 7f), was comparable to the Se-Se stretching mode of pyrite-type MnSe$_2$ (267 cm$^{-1}$ with $d_{Se-Se}$ = 2.38 Å)$^{50}$. However, like the oxysulfide, the collapsed oxyselenide was not Cu/Li-free; ICP-MS suggested Sr/Mn/Cu/Li = 2.00/1.02/0.38/0.21 (see Supplementary Table 4) and Scanning Electron Microscopy Energy Dispersive Spectroscopy (SEM-EDS) analyses showed a heterogeneous distribution of its Cu$^{+}$ cations (Supplementary Fig. 31). These results suggest that these complex intergrowth structures are inevitable products of this chemistry.

In conclusion we have demonstrated multistep topochemistry of the oxide chalcogenides Sr$_2$MnO$_2$Cu$_{1.5}$S$_2$ and Sr$_2$MnO$_2$Cu$_{1.5}$Se$_2$ via lithiated reactive intermediates leading to conversion of their copper chalcogenide slabs into 2D arrays of chalcogen dimers [$Ch_2$]$^{2-}$ with perovskite-type Sr$_2$MnO$_2$ slabs retaining structural integrity. This is a chemical complement to electrochemical cycling previously carried out for the related oxysulfide Sr$_2$MnO$_2$Cu$_{3.5}$S$_3$$^{41}$. Accordingly, our current efforts are aimed at extending the scope of this chemical route to the whole Sr$_2$MnO$_2$Cu$_{2m-\delta}$S$_{m+1}$ ($m$ = 1–3) homologous series$^{30}$. Cu extrusion through topochemical Cu-Li exchange is widely known among metal chalcogenides but mostly studied in the context of Li-ion battery applications$^{51,52}$. The combination of this extrusion with the reaction with oxidising agents such as disulfiram, a stable covalently bonded molecule with strong chemoselectivity, may offer a useful route to convert various known chalcogenides$^{53}$ into exotic structures with chalcogen-chalcogen bonds.

This chemistry highlights redox competition between cations and anions. Mn$^{4+}$ is common in oxides such as MnO$_2$, while MnS$_2$ contains oxidised [S$_2$]$^{2-}$ species, due to Mn$^{2+/3+}$ bands lying below the top of the S 3s/3p bands$^{54}$. In Sr$_2$MnO$_2$Cu$_{1.5}$S$_2$, heteroleptic MnO$_4$S$_2$ octahedra experience oxidation of Mn during partial Cu deintercalation using I$_2$$^{15}$, but further Cu deintercalation giving the collapsed phase described here involves predominantly oxidation of chalcogen anions, so Mn$^{2+/3+}$ cation redox is in tight competition with S$^{2-/-}$ anion redox in the oxychalcogenides. This shows that redox reactivity of anions may be controlled by heteroleptic coordination in mixed-anion systems$^{55,56}$. We are currently investigating how the competition between Mn$^{2+/3+}$ and S$^{2-/-}$ redox evolves during Li (de)intercalation processes as well as the complex reaction dynamics both from thermodynamic and kinetic point of views.

The new disulfide and diselenide-containing products have the ideal composition Sr$_2$MnO$_2$Ch$_2$, but regions with such a composition are found to co-exist in complex intergrowth phases with approximated overall composition Sr$_2$MnO$_2$(Cu,Li)$_{0.5}$Ch$_2$. We modelled the presence of these residual Cu$^{+}$ and Li$^{+}$ cations using stacking faults in which monoclinic Sr$_2$MnO$_2$Ch$_2$ regions intergrow with parent-type Sr$_2$MnO$_2$Cu$_{1.5}$S$_2$ slabs, but our Rietveld analysis could not fully account for their compositional and structural heterogeneity. On the other hand, our local probe analyses using HAADF-STEM imaging and $^{7}$Li NMR revealed that substantial regions consisted of periodic intergrowths of Sr$_2$MnO$_2$Ch$_2$ and Sr$_2$MnO$_2$(Cu,Li)$_2$S$_2$-type layers. This evidence for the intergrowth structures may provide clues to the mechanism of the deintercalation reactions and suggest that targeted synthesis of perfectly periodic intergrowth phases, perhaps offering important electronic and magnetic properties, should be pursued.

## Methods
### Synthesis
Unless otherwise noted, all procedures were performed under inert atmosphere. The synthetic scheme is outlined in Fig. 1a. The parent phase Sr$_2$MnO$_2$Cu$_{1.5}$S$_2$ was prepared by the reaction between SrS (2 equivalents) (obtained by reacting SrCO$_3$ (Alfa 99.994%) with CS$_2$ (Aldrich 99%) carried by flowing argon in a tube furnace in a fume cupboard (caution: CS$_2$ is highly toxic and flammable)), MnO$_2$ (Aldrich 99.99%) (0.25 equivalents), Mn (Aldrich >99%) (0.75 equivalents) and CuO (Alfa 99.995%) (1.5 equivalents). The mixture was ground in an

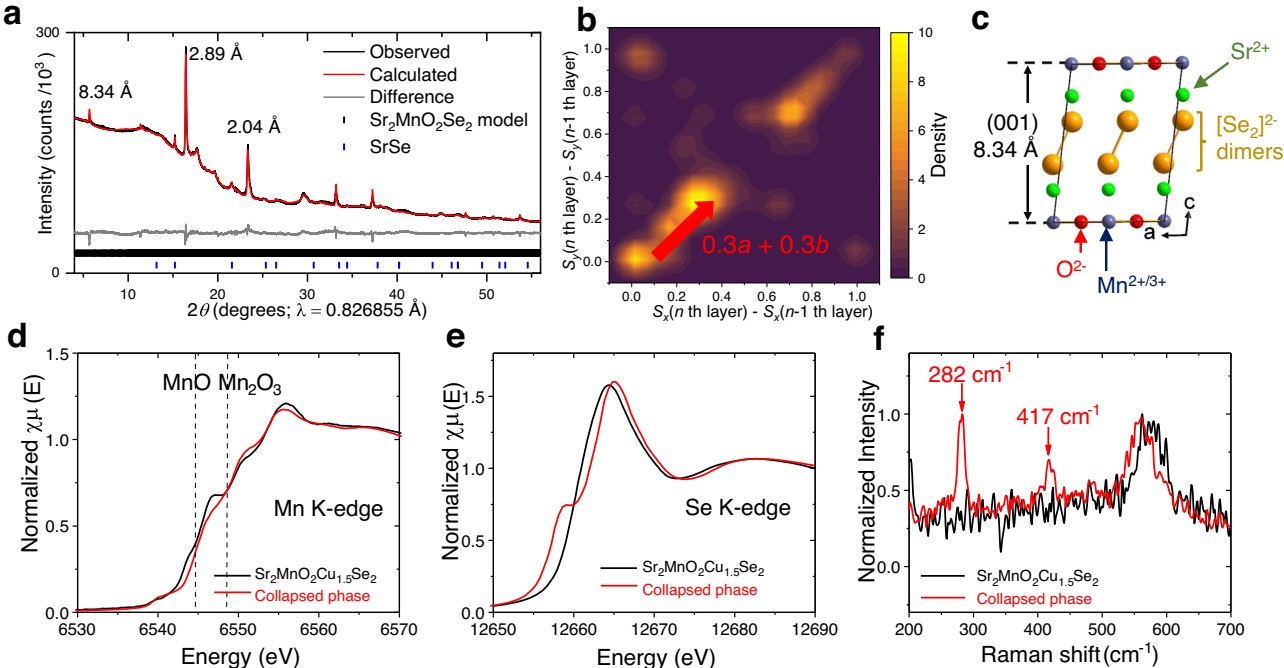

**Fig. 7 | Collapsed manganese oxyselenide and oxidation of its selenide anions.** **a** Rietveld fit to powder synchrotron PXRD data (I11, Diamond) of the collapsed oxyselenide derived from $Sr_2MnO_2Cu_{1.5}Se_2$. $R_{wp} = 1.40\%$, $\chi^2 = 4.7$. The same method described in Fig. 2b was used for the refinement. **b** 2D kernel plot representing the occurrence of $(S_x, S_y)$ values refined for sliding of each $Sr_2MnO_2Se_2$ layer. See Supplementary Fig. 29 for details. **c** The collapsed $Sr_2MnO_2Se_2$ model derived by the most frequent shift $(S_x, S_y) = (0.3, 0.3)$. **d, e** XANES spectra at Mn K-edge (**d**) and Se K-edge (**e**) were measured for the pristine $Sr_2MnO_2Cu_{1.5}Se_2$ (black) and its collapsed phase (red). Dotted lines represent the absorption edges of reference standards MnO and $Mn_2O_3$. **f** Raman spectra of $Sr_2MnO_2Cu_{1.5}Se_2$ (black) and its collapsed phase (red). The measurement was performed under air with excitation wavelength $\lambda_{ex} = 532$ nm.

agate pestle and mortar, pressed into a 13 mm pellet and loaded into alumina crucible which was sealed under vacuum in a dried silica tube. This was heated to 900 °C at 10 °C min⁻¹ and after a further 48 hours the furnace was turned off and the sample cooled at the natural rate of the furnace[15,30]. The selenide analogue $Sr_2MnO_2Cu_{1.5}Se_2$ was prepared in a similar way[47] but with the starting materials: SrO (from thermal decomposition of $SrCO_3$) (2 equivalents), Mn (Aldrich >99%) (1 equivalent), Cu powder (Alfa 99.995%) (2 equivalents) and Se (Alfa 99.999%) (2 equivalents). For the selenide the pellet was heated at 5 °C min⁻¹ to 850 °C and this was maintained for 24 h followed by natural cooling of the furnace. The initial lithiation of both $Sr_2MnO_2Cu_{1.5}S_2$ and $Sr_2MnO_2Cu_{1.5}Se_2$ was performed[21] by stirring a suspension of the ground powder with a 20-fold excess (Li: Mn ratio 20: 1) of 1.6 M $n$-BuLi in hexane at 50 °C for 4 days.

The obtained mixture $Sr_2MnO_2Li_xS_2 + y$ Cu was loaded into a Schlenk tube with 2.0 equiv. of disulfiram which was then dissolved in anhydrous dimethylformamide (DMF) so as to make the concentration of disulfiram around 0.18 M. Better selectivity toward Cu dissolution was observed when small portions of DMF were added at the beginning, and the rest of the solvent was not added until the exothermic process ceased. The suspension was stirred at room temperature overnight, followed by washing with anhydrous tetrahydrofuran (THF) that was repeated until the filtrate became colourless. The filtrate was dried under vacuum and subject to further treatment with $n$-BuLi in hexane at 50 °C[21] and subsequent Cu dissolution with disulfiram once again. In the final step, the reaction mixture was combined with 1.0 equivalent of disulfiram which was then dissolved in anhydrous DMF and stirred at 80 °C for *ca.* 12 h. After washing with anhydrous THF three times, the reaction mixture was dried under vacuum to afford a black powder with the PXRD pattern displayed in Fig. 2a (Mass yield: 70%). The oxyselenide $Sr_2MnO_2Cu_{1.5}Se_2$ was subject to the same protocol of Cu-Li exchange and subsequent treatments with disulfiram at ambient temperature. The final step of Fig. 1a (i.e. treatment with disulfiram at 80 °C) was not required for the oxyselenide. The reaction with disulfiram at ambient temperature produced the pattern shown in Fig. 7a and further reaction at an elevated temperature of 80 °C did not lead to any change in the pattern. The syntheses were successfully carried out up to 2 g scale, but it must be kept in mind that the lithiation step uses large excess of the pyrophoric organolithium reagent at elevated temperature. The reaction therefore must be done with appropriate safety precautions[57].

## X-ray and neutron powder diffraction measurements

Structural analyses of the deintercalated oxysulfide and oxyselenide phases were performed using synchrotron PXRD acquired at beamline I11 (Diamond Light Source, Harwell, UK)[58] with a Si-calibrated incident wavelength of 0.82686(7) Å. The samples were sealed in 0.5 mm diameter capillaries under Argon. In addition to these measurements at room temperature, the synchrotron PXRD was measured also on heating from 115 °C to 400 °C at a rate of 6 °C min⁻¹ with a PXRD pattern collected every 3 °C (Supplementary Fig. 6). The structure analysis of the collapsed oxysulfide phase was complemented by collecting time-of-flight powder neutron diffraction (PND) data on the medium-resolution, high-flux instrument POLARIS[59] at the ISIS pulsed neutron facility (Rutherford Appleton Laboratory, Harwell, UK). A separate, large (~1.5 g) sample was loaded into a 6 mm diameter thin-walled vanadium can in a glove box and sealed with indium wire. Diffraction data were acquired at room temperature (~293 K) for 400 µAh proton beam current onto the ISIS target (~2 hours of beamtime). Other synthetic intermediates, as well as the products of chemical lithiation of the collapsed phase were examined using a Bruker D8 Advance Eco X-ray diffractometer (Bragg-Brentano geometry, $\theta$–$2\theta$) operated at 40 kV and 25 mA with Cu $K_\alpha$ radiation. For these laboratory PXRD measurements, the powder samples were sprinkled on a Borosilicate cover glass using a minimal amount of Dow Corning® high-vacuum silicone grease as an adhesive and these were mounted in an aluminium gas-tight sample holder under inert atmosphere. This setup

resulted in tiny peaks from elemental Si and Al in the PXRD patterns. All of these diffraction patterns were analysed by Rietveld refinements using the TOPAS Academic software (Version 6). The details of refinements and structure modelling were described in the Supplementary Information where further discussion of the analysis of stacking faults can be found (Supplementary Note 1).

## Magnetometry

All measurements were carried out using a Quantum Design MPMS-3 Superconducting Quantum Interference Device (SQUID) magnetometer with powder samples (sample mass: 10-20 mg, accurately measured in each case) contained in gelatine capsules. Firstly, the sample was cooled down to 2 K under zero applied field: zero-field-cooled (ZFC) and Direct Current (DC) susceptibility was measured under an applied field of 100 Oe of while sweeping temperature to 300 K. Then the sample was cooled back down maintaining the applied field of 100 Oe and measured again on sweeping the temperature to 300 K (field-cooled (FC) measurement). The sample was cooled down once again to 2 K under an applied field of 50 kOe (= 5 T) and the Magnetisation versus magnetic field isotherm was recorded by sweeping the field in steps down to −5 T and back to +5 T.

## Scanning electron microscopy (SEM) measurements

All measurements were carried out in the David Cockayne centre for Electron Microscopy, University of Oxford, using a Zeiss EVO MA10 microscope equipped with an Oxford Instruments X-act energy dispersive X-ray spectroscopy (EDS) detector. Powder samples were mounted on adhesive carbon tape and then surfaces of the samples were coated with 6 nm carbon layers using a Leica ACE600 Coater. Specimens were prepared under inert atmosphere and stored in a polythene bag until the moment of their use, but brief exposure of samples to air was inevitable during the quick transfer of specimens to the carbon coater and to the sample chamber of the SEM, resulting in possible degradation of air-sensitive samples, in particular those obtained after chemical lithiation. For imaging, secondary electron (SE) images were acquired and combined with EDS mapping at an operating voltage of 20 kV (sulfide systems) or 10 kV (selenide systems) with a probe current of 1.0 nA. Backscattered electron (BSE) images were equally closely examined but no further information about compositional heterogeneity was obtained. EDS maps were processed using the Oxford Instruments Aztec software for initial assessments of spatial distribution of each element as well as the presence of impurity phases. This was followed by EDS measurements at individual points or at series of points on predefined lines (using linescan mode) in order to obtain quantitative views of their compositions.

## X-ray absorption near-edge spectroscopy (XANES)

All XANES spectra were collected at beamline B18 (Diamond Light Source, Harwell, UK) equipped with a Si (111) double-crystal monochromator. The measurements were performed in transmission mode for Mn and Se K-edges and in fluorescence mode for the S K-edge. For measurements in transmission mode, 10–20 mg of the powder samples were diluted in $ca.$ 100 mg of dried microcrystalline cellulose, pressed into 13 mm pellets, and enclosed in aluminised polythene bags under inert atmosphere. For measurements in fluorescence mode, powder samples were sprinkled on carbon tapes and samples transferred from the preparation glove box to a He-filled experimental chamber using a vacuum suitcase system specifically designed for the B18 tender-X-ray end-station. For Mn and Se K edges, samples were enclosed in aluminised polythene bags under inert atmosphere, and in both cases, their spectra were recorded without opening the polythene bags to prevent air exposure. Mn and Se K-edge spectra were calibrated respectively against the Mn K-edge and Pt $L_2$-edge of elemental metal foils placed in the reference channel. Data processing, calibration and normalisation were carried out using the Athena software package[60].

## Raman spectroscopy

Raman spectra were recorded using a ThermoFisher Scientific DXR™3 SmartRaman Spectrometer equipped with 532 and 785 nm lasers. Laser power was fixed to 100 mW and spectra were obtained in the range of 3500 $cm^{-1}$ to 100 $cm^{-1}$. The powder sample was packed in the sample holder and directly illuminated by a 532 nm excitation beam for 16 periods of 4 sec after acquisition of the background signals. To check effects of sample atmosphere and excitation wavelength, the measurement was also carried out for the oxysulfide samples sealed in silica capillaries under inert atmosphere and exposed to a 785 nm excitation beam.

## Inductively coupled plasma-mass spectrometry (ICP-MS)

The specimens were prepared by dissolving $ca.$ 5 mg of each powder sample in 20 ml of $HNO_3$ (conc.): HCl (12 M) = 19: 1 (v/v) and then diluting them 50 times with deionised water so as to make 2 vol% solution of the original concentrated acids. The prepared sample solutions were stored in 15 ml centrifuge vials before use. A Perkin-nElmer NexION 350D ICP-MS at the Department of Earth Sciences, University of Oxford, was used to determine the concentrations of Li, Mn, Cu and Sr (See Supplementary Tables 3 and 4). An Elemental Scientific prepFAST M5 autosampler and autodiluter was used to dilute each element within an appropriate range of the calibration standards. The elements In, Re and Rh were also added to each blank, standard and sample as internal standards.

## Transmission electron microscopy (TEM) measurements

All the measurements were carried out for Sample 1 (bulk composition estimated by ICP-MS: $Sr_{2.00(2)}Mn_{1.07(2)}O_2Cu_{0.35(0)}Li_{0.13(0)}S_2$) of the deintercalated sulphide phase. A specimen for the TEM study was prepared by grinding the material under ethanol and depositing a few drops of the suspension onto a copper TEM grid covered by a holey carbon layer. The specimen was prepared in air. Three-dimensional electron diffraction (3D ED) data were acquired on an FEI Tecnai G2 transmission electron microscope equipped with Gatan charge-coupled device (CCD) camera operated at 200 kV. Each raw electron diffraction pattern was recorded at intervals of 1° tilt within the tilting range of −65° to 70° and these were processed by means of the PETS software in order to reconstruct the main zone sections[61].

High-angle annular dark-field (HAADF) scanning transmission electron microscopy (STEM) images were acquired using a FEI Titan 80−300 Cubed microscope operated at 300 kV. The simulated HAADF-STEM images were calculated using the QSTEM 2.5 software[62].

## Electrochemical study

Electrochemical lithium (de)intercalation was examined for Sample 1 of the collapsed sulfide phase using coin cells (CR2032, Cambridge Energy Solutions) assembled in an Ar-filled glove box ($H_2O$, $O_2$ contents <1 ppm). The deintercalated sulfide phase was mixed with Super P Carbon (Timcal) and polyvinylidene fluoride (PVDF) binder (weight ratio: 80:10:10) to prepare the cathode. A coin cell was assembled with cathode, borosilicate glass fibre separator (Whatman, 15 mm diameter) soaked with 75 µl electrolyte, and Li counter electrode (diameter 13 mm). 1 M $LiPF_6$ in ethylene carbonate (EC):dimethyl carbonate (DMC) (1:1) was used as the electrolyte. Galvanostatic (dis)charge was carried out at room temperature with a Lanhe battery cycler (Wuhan Land Electronics Co. Ltd.) at C/10 rate, where C is defined as the theoretical capacity of the compound and C/10 means fully charging or discharging over 10 hours. Prior to ex situ measurements, batteries were disassembled inside the Ar glove box and the electrode mixture was rinsed three times with DMC and dried in the glove box ante-chamber under vacuum for 30 mins. The resulting samples were recovered in powder form and packed in rotors for ex situ NMR measurements (Supplementary Fig. 26).

## Nuclear magnetic resonance (NMR) spectroscopy

[7]Li magic-angle spinning (MAS) NMR experiments were performed with a Bruker Avance 200 MHz (4.7 T) spectrometer operating at a [7]Li Larmor frequency of 77.8 MHz at room temperature with a single channel 1.3 mm Bruker probe. Samples were packed into a 1.3 mm $ZrO_2$ rotor inside the glove box and spun at speeds between 35–60 KHz. A rotor-synchronised Hahn echo sequence (90°-τ–180°-τ-acquisition) with a π/2 pulse length of 1.0 µs and recycle delay of 0.1 s was used. The spectra were referenced to a standard LiF at −1 ppm. Bruker Topspin (version 4.0.7) was used for raw data processing.

## Data availability

The data supporting the findings of this study are displayed within the article and its Supplementary Information files. Neutron Powder Diffraction data are available from https://doi.org/10.5286/ISIS.E.RB2190055-1. Further data are available from the corresponding author upon request.

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

## Acknowledgements

The project was supported by UK EPSRC grants (EP/R042594/1 (SJC), EP/P018874/1 (SJC) and EP/T027991/1 (SJC)). For the purpose of open access, the author has applied a Creative Commons Attribution (CC BY) licence to any Author Accepted Manuscript version arising. The XRD and XAS measurements were carried out as a part of Diamond Light Source Block Allocations CY25166 on I11 and SP14239 on B18, respectively. Experiments at the ISIS Neutron and Muon Source were supported by a beamtime allocation XB2190055 from the Science and Technology Facilities Council. Data is available here: https://doi.org/10.5286/ISIS.E.RB2190055-1. S.G. was supported by a University of Oxford Clarendon scholarship. S.S. thanks Mr. E. Yang and Dr. J. Holter from the University of Oxford for their training on Raman and SEM facilities, respectively.

## Author contributions

S.S. and S.J.Cl. conceived the idea and designed the experiments. S.S. and S.G. carried out the synthesis. S.J.Ca. performed stacking fault analyses. S.D. and C.P.G. conducted the electrochemical studies and $^7$Li NMR analyses. M.B., D.V. and J.H. carried out 3D electron diffraction studies and HAADF-STEM imaging with TEM. G.C., R.I.S. and P.H. collected XANES, neutron diffraction and ICP-MS data, respectively. S.S. wrote the initial draft and all authors discussed the results and contributed to the manuscript.

## Competing interests

The authors declare that none of them have any competing financial, personal or professional relationships with individuals or institutions that could be perceived to directly undermine the objectivity, integrity, and value of the work in this article, or could be seen as having an influence on the judgments and actions of the authors with regard to objective data presentation, analysis, and interpretation.
