## [Peer Review File · Nature Communications]

REVIEWER COMMENTS

Reviewer #1 (Remarks to the Author):

Review report for NCOMMS-22-52782-T

Anionic redox-based topochemical reactions have attracted attention in recent years due to their ability to improve the performance of battery materials and synthesize new inorganic compounds containing anionic molecules. In this study, the authors focused on $\text{Sr}_2\text{MnO}_2\text{Cu}_{1.5}\text{Ch}_2$ ($\text{Ch} = \text{S}, \text{Se}$) with the alternative stacks of Sr_2MnO_2 and $\text{Cu}_{1.5}\text{Ch}_2$ layers, and in particular, Cu deintercalation in $\text{Cu}_{1.5}\text{Ch}_2$ layers from topochemical reactions utilizing the anionic redox of the formation/deformation of $(\text{Ch}_2)_2^-$ dimers in conjunction with the synthesis of a new compound. Attempts have been made to remove Cu in $\text{Sr}_2\text{MnO}_2\text{Cu}_{1.5}\text{Ch}_2$ but conventional reaction methods using e.g., I_2 and Br_2 have not been able to extract Cu sufficiently. Therefore, research from a new viewpoint was necessary. In this study, the authors proposed the use of the intermediate phase of the Li^+/Cu^+ exchanged compound to lower the activation barrier, and also selectively removed the precipitated Cu with disulfiram. By using this Li-exchanged intermediate phase as a precursor for the topochemical reaction, they demonstrated the synthesis of a novel layered compound containing $(\text{Ch}_2)_2^-$ dimers.

This research is interesting in view of the development of a new reaction route and synthesis of a compound previously inaccessible, through an intermediate phase. However, this manuscript focused on somewhat technical matters, including the reaction and the structural analysis of the obtained material, and it seemed unclear what was important as scientific knowledge. For example, it may be found that it is better to use disulfiram while probably considering the HASB theory, etc., but it seemed unclear in this manuscript from what point of view the chemically selective reagent was chosen. The generality and scalability of the chemical reactions dealt with in this study also seemed unclear.

The material obtained in this study may be a novel layered compound containing $(\text{S}_2)_2^-$ (or $(\text{Se}_2)_2^-$) dimers. However, at present, stacking faults and intergrowth structures exist randomly, i.e., it could be an impure mixture. In addition, even if Li can be electrochemically inserted, there is a limit to sufficiently removing Li again, resulting in a limit to the development of applications such as batteries.

In this research, it is important to focus on anionic redox and develop multi-step topochemical reactions. It was thus thought that the manuscript would be better if there was a clear description of how the strategy for the synthesis of new compounds and an understanding science behind this.

The followings are other points that seemed unclear:

1. In the middle of the multi-step reaction (STEP2,4 in Fig.1), disulfiram was used to remove the precipitated Cu. Is there another way to remove Cu? Also, in view of the synthesis of new materials using anion redox, for example, is it not possible to perform the reaction of STEP 5 after STEP 1? Besides, is it possible to directly react $\text{Sr}_2\text{MnO}_2\text{Cu}_{1.5}\text{Ch}_2$ with disulfiram without going through an intermediate phase?

2. It is stated that the obtained "collapsed" phase returns to the original precursor with the $I4/m$ structure after electrochemical lithium intercalation or reaction with n-BuLi (p.13, lines 11-16). Do the stacking faults found in the "collapsed" phase (as seen in e.g., Fig.4) disappear and return to the original structure after these reactions?

3. Is the obtained "collapsed" phase stable in the atmosphere? Is there any possibility that the composition or structure will change over time?

4. What is the reason why the novel layered compound $\text{Sr}_2\text{MnO}_2\text{S}_2$ containing $(\text{S}_2)^{2-}$ dimers and its intergrowth structure could not be synthesized in their single phases? For example, is it because the structural stability of $\text{Sr}_2\text{MnO}_2\text{S}_2$, intergrowth structure compounds, and precursor $\text{Sr}_2\text{MnO}_2\text{Li}_{1.9}\text{S}_2$ are energetically competing? It may be considered that by calculating the total energy and phonon dispersion of these compounds, it would be possible to evaluate the thermodynamic stability of these compounds and clarify this reason.

Minor point

I thought it might be good to move the sentence about the details of the Rietveld analysis described in the second paragraph on page 13 to Supporting Information. The stacking faults and random distribution of intergrowth structures of the "collapsed" phase discussed in this paragraph seemed to be already clarified from experiments of STEM and EDX elemental mapping. Therefore, the reason for repeating the Rietveld analysis seemed unclear. I thought that the authors had a great effort for the Rietveld analysis with the periodic structural model and carefully performed the analysis. However, in the end, the authors concluded that the structural model considered here could not reasonably reproduce the diffraction peak intensity. Therefore, it seemed to me that the claim from the Rietveld analysis here is ambiguous.

(END of comments)

Reviewer #2 (Remarks to the Author):

This is a high-quality work about topochemical reaction of layered oxychalcogenides. The author reported multi-step Cu-deintercalation of $\text{Sr}_2\text{MnO}_2\text{Cu}_{1.5}\text{Ch}_2$ yielding the collapsed phase with Ch_2 dimers. The (almost) full deintercalation of Cu from $\text{Sr}_2\text{MnO}_2\text{Cu}_{1.5}\text{Ch}_2$ phase is the first case, but a similar concept was already reported by the authors in the other layered oxychalcogenides (e.g. ref 14; $\text{La}_2\text{O}_2\text{Cu}_2\text{S}_2$). The impressive point in this work is the controlled chemical reactions: They used an organic reagent, disulfiram, which seems to work as a chemoselective agent toward Cu. Their characterization of the collapsed phase is well done in that they used several methods to determine the structural and chemical states. Especially, they carried out advanced Rietveld analysis to fit the XRD pattern with defects or stacking faults.

I consider that this work is important for the fundamental interests of solid-state chemistry fields, and is worth publishing in Nat. Commun.

Additional comment:

I feel that the five-step-reaction is one of the most important part of this work. But there are no figures to see how XRD patterns changes in each reaction (There are some XRD patterns in supporting information but they are not whole reactions). I recommend that the authors put the five XRD patterns in the main figure.

Reviewer #3 (Remarks to the Author):

Sasaki and coworkers report on the topochemical manipulation of a layered mixed-metal oxychalcogenide compounds. Sequential reaction steps involving copper extraction and lithium intercalation/deintercalation result in a series of topochemically related intermediates, culminating in a "collapsed" structure with new sulfur/selenium dimers. This result is exciting in that it shows the rigorous extraction of intermediate layers can be carried out while still retaining key structural features (MnO layers) of the parent compound. The researchers are thorough in their treatment of this system, including detailed X-ray, neutron, and electron diffraction studies. The electron diffraction studies are especially illuminating as to the concurrent complexity and beauty of the final products.

This research is of general interest but will be especially appealing to solid state chemists, researchers interested in topochemistry, and those focused on complex redox processes in energy storage materials.

The research is appropriate for Nature Communications and can be published without revision.

NCOMMS-22-52782-T

**Anion Redox as a Means to Derive Layered Manganese
Oxychalcogenides with Exotic Intergrowth Structures**

S. Sasaki *et al.*

Response to Reviews.

Reviewer #1 (Remarks to the Author):

Review report for NCOMMS-22-52782-T

Anionic redox-based topochemical reactions have attracted attention in recent years due to their ability to improve the performance of battery materials and synthesize new inorganic compounds containing anionic molecules. In this study, the authors focused on $\text{Sr}_2\text{MnO}_2\text{Cu}_{1.5}\text{Ch}_2$ ($\text{Ch} = \text{S}, \text{Se}$) with the alternative stacks of Sr_2MnO_2 and $\text{Cu}_{1.5}\text{Ch}_2$ layers, and in particular, Cu deintercalation in $\text{Cu}_{1.5}\text{Ch}_2$ layers from topochemical reactions utilizing the anionic redox of the formation/deformation of $(\text{Ch}_2)^{2-}$ dimers in conjunction with the synthesis of a new compound. Attempts have been made to remove Cu in $\text{Sr}_2\text{MnO}_2\text{Cu}_{1.5}\text{Ch}_2$ but conventional reaction methods using e.g., I_2 and Br_2 have not been able to extract Cu sufficiently. Therefore, research from a new viewpoint was necessary. In this study, the authors proposed the use of the intermediate phase of the Li^+/Cu^+ exchanged compound to lower the activation barrier, and also selectively removed the precipitated Cu with disulfiram. By using this Li-exchanged intermediate phase as a precursor for the topochemical reaction, they demonstrated the synthesis of a novel layered compound containing $(\text{Ch}_2)^{2-}$ dimers.

This research is interesting in view of the development of a new reaction route and synthesis of a compound previously inaccessible, through an intermediate phase. However, this manuscript focused on somewhat technical matters, including the reaction and the structural analysis of the obtained material, and it seemed unclear what was important as scientific knowledge. For example, it may be found that it is better to use disulfiram while probably considering the HASB theory, etc., but it seemed unclear in this manuscript from what point of view the chemically selective reagent was chosen. The generality and scalability of the chemical reactions dealt with in this study also seemed unclear.

The material obtained in this study may be a novel layered compound containing $(\text{S}_2)^{2-}$ (or $(\text{Se}_2)^{2-}$) dimers. However, at present, stacking faults and intergrowth structures exist randomly, i.e., it could be an impure mixture. In addition, even if Li can be electrochemically inserted, there is a limit to sufficiently removing Li again, resulting in a limit to the development of applications such as batteries.

In this research, it is important to focus on anionic redox and develop multi-step topochemical reactions. It was thus thought that the manuscript would be better if there was a clear description of how the strategy for the synthesis of new compounds and an understanding science behind this.

Response:

We thank the referee for these constructive suggestions. We understood the overall concern from the referee that the manuscript focused on technical matters, and therefore further

discussion was added to elaborate on the scientific ideas behind our synthetic route, perspectives for further applications & challenges to be tackled. The following are a point-by-point responses to the referee's major concerns.

- "...it may be found that it is better to use disulfiram while probably considering the HASB theory, etc., but it seemed unclear in this manuscript from what point of view the chemically selective reagent was chosen"
→ As the referee suggested, our motivation for lithiation is based on HSAB theory. Small, hard-acidic Li^+ cations are less compatible with tetrahedral sites surrounded by soft-basic S^{2-} anions than more soft-acidic Cu^+ cations. Accordingly, we hypothesized that Li^+ could be removed more easily from the host oxysulfides.

To clarify this, we added the phrase in Page 6:

"Compared to the soft-acidic Cu^+ intercalants, the smaller, hard-acidic Li^+ cations can be removed more easily from the soft-basic sulfide layers."

As for the choice of disulfiram, its primary reason is the excellent chemoselectivity toward Cu^0 dissolution as already mentioned in the manuscript. Its reduced form, dithiocarbamate, is widely known as a chelating agent to extract polyvalent transition metals like Cu^{2+} (See for example: Gallagher, W. P. et al. *Org. Process Res. Dev.* **19**, 1369–1373 (2015)). The below figure is the XRD pattern of the by-products that we collected from the supernatant solution of the Step 2 reaction. Our cursory Rietveld fit used only the model of $\text{Cu(II)bis}(N,N\text{-diethyldithiocarbamate})$ but qualitatively explained most of its diffraction peaks. This reflects the strong preference of disulfiram to form complexes with Cu over Li. As we already presented in Fig. S3, less chemoselective reagents such as I_2 gave the completely opposite results through Li^+ deintercalation and Cu re-intercalation into the host lattice.

Fig. A. Rietveld fit of the XRD pattern obtained after drying the washed supernatant solution of Step 2 reaction mixture. The $\text{Cu(II)bis}(N,N\text{-diethyldithiocarbamate})$ model was retrieved from CCDC 1123544 (ref: M.Bonamico et al. *Acta Crystallo.* **19**, 886 (1965).)

Another reason to choose disulfiram is its molecular structure. Disulfiram has a stable, covalently bonded molecular skeleton made of C, N, S and non-acidic protons. As noted already in the introduction, oxidizing agents based on halogens and nitronium ions (e.g. I_2 at above 0°C , Br_2 , NO_2BF_4) tend to decompose $\text{Sr}_2\text{MnO}_2\text{Cu}_{1.5}\text{S}_2$ (see Blandy et al. *APL Mater.* **3**, 041520 (2015) for details). This was the case even when the ionic liquid $[\text{Bmim}]^+[\text{FeCl}_4]^-$ was used as a Lewis-acidic, mild oxidizing agent (ref: Amarasekara, A. S. *Chem. Rev.* **116**, 6133-6183 (2016)), which ended up with decomposition of the host oxysulfide into SrCl_2 etc. Besides halogens, we also wanted to avoid reagents containing acidic protons that often destroy Li-S compounds. That was the reason why we did not use other well-known metal chelating agents such as

EDTA and Salicylaldoxime. So far, we regard disulfiram as the best reagent to synthesize the collapsed phase but all of the above criteria may be applied to other reagents. Especially, further optimization of its molecular structure would be a useful avenue for future research rather than simply using commercially available reagents.

Based on the above discussion, we added the following sentences in page 7:

“Equally, one must properly design oxidizing agents so that they selectively dissolve elemental Cu without affecting the rest of the host framework. For example, the reaction of the Step 1 product with I₂ in acetonitrile, a conventional oxidant for chemical deintercalation, simply restored the parent phase through Li deintercalation and almost complete Cu reinsertion without any Cu dissolution (Fig. S3). The soft-basic, bidentate dithiocarbamate ligands have a strong preference to form complexes with polyvalent transition metal cations over monovalent alkali metal cations²⁵. In addition, disulfiram’s covalently bonded skeleton made of C, N, S and non-acidic protons are less likely to destroy the host lattice compared to reagents made of halogens, nitronium or acidic protons.”

Ref 25: Gallagher, W. P., Vo, A. Dithiocarbamates: Reagents for the Removal of Transition Metals from Organic Reaction Media. *Org. Process Res. Dev.* 19, 1369–1373 (2015).

- “The generality and scalability of the chemical reactions dealt with in this study also seemed unclear.”

→ In our study, we have successfully applied the chemical route not only to the sulfide but also the selenide. As for future perspective, we currently confirmed that the similar chemical route could be applied to the whole Sr₂MnO₂Cu_{2m-δ}S_{m+1} (m = 1-3) homologous series. There are also many other known compounds made of anti-fluorite type copper chalcogenide layers. Disulfiram may serve as effective reagents to remove copper, either directly from the host lattice triggering collapses of the layers (although that is not successful in this case – see below), or indirectly via the Cu-Li exchange steps used in this case.

To discuss generality of the reactions, the following sentences were added in page 18:

“Accordingly, our current efforts are aimed at extending the scope of this chemical route to the whole Sr₂MnO₂Cu_{2m-δ}S_{m+1} (m = 1-3) homologous series³⁰. Cu extrusion through topochemical Cu-Li exchange is widely known among metal chalcogenides but mostly studied in the context of Li-ion battery applications⁵¹⁻⁵². The combination of this extrusion with the reaction with oxidizing agents such as disulfiram, a stable covalently bonded molecule with strong chemoselectivity, may offer a novel route to convert various known chalcogenides⁵³ into exotic structures with chalcogen-chalcogen bonds.”

Ref 30: Gal, Z. A., Rutt, O. J., Smura, C. F., Overton, T. P., Barrier, N., Clarke, S. J. & Hadermann, J. Structural Chemistry and Metamagnetism of an Homologous Series of Layered Manganese Oxyulfides. *J. Am. Chem. Soc.* **128**, 8530-8540 (2006).

Ref 51: Morcrette, M. et al. A reversible copper extrusion–insertion electrode for rechargeable Li batteries. *Nature Mater* **2**, 755–761 (2003).

Ref 52: Bodenez, V. et al. Copper Extrusion/Reinjection in Cu-Based Thiospinels by Electrochemical and Chemical Routes. *Chem. Mater.* **18**, 4278-4287 (2006).

Ref. 53: Clarke, S. J. et al. Structures, Physical Properties, and Chemistry of Layered Oxychalcogenides and Oxypnictides. *Inorg. Chem.* **47**, 8473–8486 (2008).

As for scalability, we added the short note at the end of the method part (Page 20):

“The syntheses were successfully carried out up to 2g scale, but it must be kept in mind that the lithiation step uses large excess of the pyrophoric organolithium reagent at

*elevated temperature. The reaction therefore must be done with appropriate safety precautions.*⁵⁷

Ref 57: Schwindeman, J. A., Woltermann, C. J., Letchford, R. J. Safe handling of organolithium compounds in the laboratory. *Chem. Health Saf.* **9**, 6–11 (2002).

- “...even if Li can be electrochemically inserted, there is a limit to sufficiently removing Li again, resulting in a limit to the development of applications such as batteries.”
→ We agree with the referee on the limited electrochemical cyclability of the collapsed phase. As shown in Fig. 4d, electrochemical Li deintercalation (i.e. Charge step) showed large hysteresis in voltage. This path hysteresis originates from multiple factors including 1) poor electronic conductivity 2) difference in the activation barriers associated with Li cation migration vs and the structural rearrangements associated with S-S bond formation. From our earlier studies it was clear that the faster migration of Li vs Cu in this class of structure during both discharge and charge, led to considerable hysteresis, anion redox being enabled because that was kinetically easier than Cu reinsertion (ref. S. Dey. et. al. *Chem. Mater.* **2021**, *33*, 3989–4005). Qualitatively similar path hysteresis has also been observed among Li-excess sulphide and oxide-based cathodes which show anion redox, however, the mechanisms of these kinetic asymmetries are subject to intense debates (See e.g. Li, B., Sougrati et al. *Nat. Chem.* **13**, 1070–1080 (2021), Van der Ven, Anton, et al. "Hysteresis in electrochemical systems." *Battery Energy 1.2* (2022): 20210017). Underpinning this class of hysteresis is a different structural pathway travelled on discharge and charge. These reports also show that the voltage hysteresis could be tuned including the proper modification of electronic and ionic structural parameters. Therefore, we believe it is too early to dismiss this class of materials from the standpoint of battery application. For example, the Cu-free collapsed structure may circumvent the Cu dendrite that induced short circuits as well as Cu dissolution problems experienced in previous battery studies based on Cu extrusion (P. Poizot et al. *Electrochem. Solid State Lett.* **2005**, *8*, A184). In future the in-depth analyses of electrochemical behaviour may provide us a guide to circumvent the potential barrier, rendering the collapsed phase (or its variants) promising as cathode materials.

We are currently working to improve our understanding of the electrochemical behaviours of this phase. Our preliminary studies suggest that the reduction of this collapsed phase follows a straightforward intercalation mechanism, unlike the Li-Cu exchange seen for Cu analogues. The kinetic barrier for S²⁻ oxidation to form S-S bonds appears to cause the voltage hysteresis. Since this analysis is ongoing we thus added a short discussion concerning the challenges (See page 12):

“The subsequent charging step in Fig. 4d exhibited a large voltage hysteresis and a capacity of only 60% of that obtained during discharge. Such a path hysteresis is common among electrochemistry involving anion redox, with long-standing debates about its mechanisms^{12,35-38}. One explanation ascribed energy inefficiency during anionic oxidation to sluggish structural distortions¹⁷, which sometimes took days or weeks to reach equilibrium³⁹. Similarly, the voltage hysteresis in the current system can be ascribed to the large structural rearrangement associated with the formation of the collapsed phase and its likely high associated activation energy. The presence of two electrochemically active centers, i.e. Mn^{2+/3+} and S^{2-/-} redox, potentially enables charge and discharge processes to take different pathways of structure transformation, leading to a voltage hysteresis^{38,40}. The presence of residual Cu may also play a role in this

hysteresis, as seen previously in the Cu-system⁴⁶. In any case, such kinetic barriers also hint at why the chemical deintercalation at Step 6 (Fig. 1a) could not remove all Cu/Li cations from the final product. Further in-depth analyses of its reaction dynamics are currently ongoing, employing in- or ex-situ spectroscopic and diffraction techniques.”

Ref 12: Hansen, C. J., Zak, J. J., Martinolich, A. J. *et al.* Multielectron, Cation and Anion Redox in Lithium-Rich Iron Sulfide Cathodes. *J. Am. Chem. Soc.* **142**, 6737-6749 (2020).

Ref 17: Assat, G., Glazer, S. L., Delacourt, C. & Tarascon J.-M. Probing the thermal effects of voltage hysteresis in anionic redox-based lithium-rich cathodes using isothermal calorimetry. *Nature Energy* **4**, 647-656 (2019).

Ref 35: Gent, W. E. *et al.* Coupling between oxygen redox and cation migration explains unusual electrochemistry in lithium-rich layered oxides. *Nat. Commun.* **8**, 2091 (2017).

Ref 36: Nagarajan S., Hwang S., Balasubramanian M., Thangavel N. K. & Arava L. M. R. Mixed cationic and anionic redox in Ni and Co free chalcogen-based cathode chemistry for Li-ion batteries. *J. Am. Chem. Soc.* **143**, 15732-15744 (2021).

Ref 37 : Jacquet, Q., Ladecola, A., Saubanère, M., Li, H., Berg, E. J., Rouse, G., Cabana, J., Doublet, M.-L. & Tarascon, J.-M. Charge transfer band gap as an indicator of hysteresis in Li-disordered rock salt cathodes for Li-ion batteries. *J. Am. Chem. Soc.* **29**, 11452-11464 (2019).

Ref 38: Basse, E. N., Reeves, P. J., Jones, M. A., Lee, J., Seymour, I. D., Cibir, G. & Grey, C. P. Structural Origins of Voltage Hysteresis in the Na-Ion Cathode $P2\text{-Na}_{0.67}[\text{Mg}_{0.28}\text{Mn}_{0.72}]\text{O}_2$: A Combined Spectroscopic and Density Functional Theory Study. *Chem. Mater.* **33**, 4890-4906 (2021).

Ref 39: Li, B., Sougrati, M.T., Rouse, G. *et al.* Correlating ligand-to-metal charge transfer with voltage hysteresis in a Li-rich rock-salt compound exhibiting anionic redox. *Nat. Chem.* **13**, 1070-1080 (2021).

Ref 40: Chang, D., Huo, H., Johnston, K. E., Ménétrier, M., Monconduit, L., Grey, C. P. & Van der Ven, A. Elucidating the origins of phase transformation hysteresis during electrochemical cycling of Li-Sb electrodes. *J. Mater. Chem. A* **3**, 18928-18943 (2015)

Ref 46 : Dey, S., Zeng, D., Adamson, P., Cabana, J., Indris, S., Lu, J., Clarke, S. J. & Grey, C. P. Structural Evolution of Layered Manganese Oxysulfides during Reversible Electrochemical Lithium Insertion and Copper Extrusion. *Chem. Mater.* **33**, 3989-4005 (2021).

The followings are other points that seemed unclear:

1. In the middle of the multi-step reaction (STEP2,4 in Fig.1), disulfiram was used to remove the precipitated Cu. Is there another way to remove Cu? Also, in view of the synthesis of new materials using anion redox, for example, is it not possible to perform the reaction of STEP 5 after STEP 1? Besides, is it possible to directly react $\text{Sr}_2\text{MnO}_2\text{Cu}_{1.5}\text{Ch}_2$ with disulfiram without going through an intermediate phase?

Response:

- “Is there another way to remove Cu?”
→ It may be possible to remove the extruded elemental Cu^0 using the proper oxidizing agents similar to disulfiram: strong preference to chelate Cu^{2+} over Li^+ , non-acidic, non-halogenic molecules with stable covalently-bonded skeletons. We did not perform extensive screening since disulfiram has already exhibited good performance. On the other hand, it is much more rewarding to attempt to dissolve Cu at the same time with Cu-Li exchange (Step 1) since such one-pot removal of Cu from the system would significantly simplify the synthetic scheme. It is not easy to dissolve Cu under the presence of a strong reducing agent like *n*-butyllithium, but we made some attempt to dissolve Cu as organolithium cuprates by adding lithium bromides and alkyldiamines. So far these efforts haven't seen any success, but we continue to explore this possibility from various approach.

Following the query from the referee, we added the sentence in page 7:

“We also tried to dissolve Cu⁺ cations at Step 1 as organolithium cuprates²⁶ before they are reduced to elemental Cu⁰ but these attempts were so far unsuccessful. Nevertheless, the one-pot removal of Cu would significantly simplify the synthetic procedure and therefore this will be one of our next challenges in synthetic methodology.”

Ref 26: Yi, H., Yang, D., Xin, J., Qi, X., Lan, Y., Deng, Y., Pao, C.-W., Lee, J.-F., Lei, A. Unravelling the hidden link of lithium halides and application in the synthesis of organocuprates. *Nat Commun* **8**, 14794 (2017).

- “...is it not possible to perform the reaction of STEP 5 after STEP 1?”
→ The difference between Step 2 and 5 is just their reaction temperature. Therefore, “Step 5 after Step1” process must necessarily go through Step 2 reaction unless hot solution ($T = 80\text{ }^{\circ}\text{C}$) were rapidly injected into Step 1 product, which would make the exothermic reaction with disulfiram too violent to be controlled. In the case of Step 1→2→5, this route also give the XRD pattern similar to that of the proper 5-step product (Fig. 2a). However, skipping Step 3-4 would end up with increased Cu content in the final product. Our present study aimed at removing Cu as much as possible to synthesize the new compounds, and therefore we did not pursue this shortcut pathway that produces more impurity “parent-type” stacks.
- “...is it possible to directly react Sr₂MnO₂Cu_{1.5}Ch₂ with disulfiram without going through an intermediate phase?”
→ In the same way with a commonly used oxidizing reagents like I₂, disulfiram can be used for partial Cu deintercalation from Sr₂MnO₂Cu_{1.5}Ch₂ without going through a synthetic intermediate. The figure below displays PXRD patterns before and after the direct reaction of Sr₂MnO₂Cu_{1.5}S₂ with disulfiram. After the reaction at 80 °C, the diffraction peaks exhibited small but visible shifts toward higher angle. Rietveld refinement using the Sr₂MnO₂Cu_{1.5}S₂ structural model estimated the cell parameters to be $a \sim 4.00\text{ \AA}$ and $c \sim 17.06\text{ \AA}$, comparable to those for the Sr₂MnO₂Cu_{1.5-x}S₂ phase ($x < 0.17$) that we previously obtained after treatment by I₂ at 0°C (For details: Blandy J. et al. *APL Mater.* **3**, 041520 (2015)). Unlike I₂, disulfiram did not decompose the host oxysulfide even above 0 °C. We can therefore recommend disulfiram as a non-volatile, non-corrosive, and non-hygroscopic alternative to I₂ or other common reagents used for deintercalation. Nevertheless, disulfiram alone was not sufficient to escape from the potential landscape of the “parent-type” *I4/mmm* structures and to reach the collapsed phase directly in this case.

The below figure was added as Fig. S2 in the supporting information. We also added the following sentences in page 6:

“The combination of the lithiation steps and the selective dissolution of the extruded elemental Cu was essential to reach the final “collapsed” phase. We also treated the parent Sr₂MnO₂Cu_{1.5}S₂ phase with disulfiram without the lithiation step (Fig. S2), but the reaction even at elevated temperature ($T = 80\text{ }^{\circ}\text{C}$) reduced its cell parameters only to the extent comparable to our previous attempt to oxidize Sr₂MnO₂Cu_{1.5}S₂ using I₂ at 0°C,¹⁵ which deintercalated Cu by about 10%. This result suggests a large activation barrier between the parent phase and the “collapsed” phase that must be circumvented by going through reactive intermediates.”

Ref 15: Blandy, J. N., Abakumov, A. M., Christensen, K. E., Hadermann, J., Adamson, P., Cassidy, S. J., Ramos, S., Free, D. G., Cohen, H., Woodruff, D. N., Thompson, A. L. & Clarke, S. J. Soft chemical control

of the crystal and magnetic structure of a layered mixed valent manganite oxide sulfide *APL Mater.* **3**, 041520 (2015).

Fig. B. X-ray diffraction (XRD) patterns after the reaction of $\text{Sr}_2\text{MnO}_2\text{Cu}_{1.5}\text{S}_2$ with disulfiram. Zoom-in views of the laboratory powder XRD patterns of pristine $\text{Sr}_2\text{MnO}_2\text{Cu}_{1.5}\text{S}_2$ (black) and the products after its treatments with excess (6.0 equiv.) of disulfiram in DMF solution at ambient temperature (blue) and at 80 °C (red). Rietveld refinements using the respective patterns indicated that the treatment with disulfiram led to small cell contraction of the $\text{Sr}_2\text{MnO}_2\text{Cu}_{1.5}\text{S}_2$ structure model, but to the extent not greater than $\text{Sr}_2\text{MnO}_2\text{Cu}_{1.5-x}\text{S}_2$ phase ($x < 0.17$) reported by Blandy J. et al. *APL Mater.* **3**, 041520 (2015).

2. It is stated that the obtained "collapsed" phase returns to the original precursor with the $I4/mmm$ structure after electrochemical lithium intercalation or reaction with $n\text{-BuLi}$ (p.13, lines 11-16). Do the stacking faults found in the "collapsed" phase (as seen in e.g., Fig.4) disappear and return to the original structure after these reactions?

Response:

After Li intercalation using $n\text{-BuLi}$, its XRD pattern (Fig. S22) no longer exhibited severe hkl -dependent and/or anisotropic peak broadening unlike the collapsed phase. This suggested good crystallinity of the recovered $\text{Sr}_2\text{MnO}_2\text{Li}_x\text{S}_2$ phase with the parent-type $I4/mmm$ structure. More importantly, its ^7Li NMR spectra after chemical or electrochemical lithiation no longer displayed the peaks arising from the intergrowth-type stacks (Fig. 6b-c). The ^7Li NMR peak centered around 400 ppm are characteristics of Li in parent type structure, as seen earlier by our groups in a separate study (ref [34]: Indris, S. et al. *J. Am. Chem. Soc.* **128**, 13354-13355 (2006)), thus indicating that these lithiated phases were more or less free (at least on NMR) from stacking faults made of the "collapsed" type slabs.

To emphasize the point, we added the following sentences in page 16:

"The multiple high frequency peaks above around 600 ppm have disappeared after chemical or electrochemical lithiation (Fig. 6b-c). Their NMR shift could fully be explained by the parent type structure with different $\text{Mn}^{2+/3+}$ ratios showing hyperfine interaction via $\text{Mn}^{2+/3+}$ -S-Li interactions. This suggested that most of the "collapsed" type stacks were removed upon lithiation and its product no longer suffered from the severe stacking disorders."

3. Is the obtained "collapsed" phase stable in the atmosphere? Is there any possibility that the composition or structure will change over time?

Response:

Both "collapsed" oxysulfide and oxyselenide were stable under air (12-24h of air exposure) or over time (left under Argon over 3-6 month) without any sign of degradation on its diffraction pattern, morphology and reactivity toward lithiation. Air stability of the "collapsed" oxysulfide is highlighted in its formation from $\text{Sr}_2\text{MnO}_2\text{Li}_x\text{S}_2$ upon air exposure (See the below figure). Once most of Cu had been removed from the system, a part of the $\text{Sr}_2\text{MnO}_2\text{Li}_x\text{S}_2$ phase spontaneously turned into the collapsed phase with the diffraction peak at $d \sim 7.9 \text{ \AA}$ via aerial deintercalation of Li. The example shown below was for the Step 3 product $\text{Sr}_2\text{MnO}_2\text{Li}_x\text{S}_2 + y \text{ Cu}$ ($x \sim 1.9, y \sim 0.1$), but the same was observed for the pure $\text{Sr}_2\text{MnO}_2\text{Li}_x\text{S}_2$ phases (Fig. S22) that were obtained by re-lithiation of the collapsed oxysulfide using *n*-BuLi. None of these $\text{Sr}_2\text{MnO}_2\text{Li}_x\text{S}_2$ samples, regardless of Li content *x*, could remain single phase after air exposure and they were partially converted into the collapsed phase once Cu had been removed from the system and could not reintercalate to compensate aerial Li deintercalation.

Fig. C. X-ray diffraction (XRD) patterns of the lithiated Step 3 products before and after air exposure. Air exposure of the Step 3 product $\text{Sr}_2\text{MnO}_2\text{Li}_x\text{S}_2 + y \text{ Cu}$ ($x \sim 1.9, y \sim 0.1$) led to emergence of the new peak at around 7.9 \AA , indicating the formation of the collapsed phase as a minor phase.

The above figure (Fig. C) was added in the supporting information as Fig. S4. We also added the corresponding discussion in page 7 of the manuscript:

"In contrast to such inaccessibility of the collapsed phase from $\text{Sr}_2\text{MnO}_2\text{Cu}_{1.5}\text{S}_2$, this oxidized phase was stable under air and spontaneously formed from $\text{Sr}_2\text{MnO}_2\text{Li}_x\text{S}_2$ by aerial deintercalation of Li once most of Cu had been removed from the system by the disulfiram. For example, air exposure of the Step 3 product (i.e. $\text{Sr}_2\text{MnO}_2\text{Li}_x\text{S}_2 + 0.1 \text{ Cu}$) led to emergence of a small XRD peak at 7.9 \AA (Fig. S4), indicating the formation of the collapsed phase as a minor phase. Similar XRD patterns were observed also in Step 2 and 4 products (Fig. 1c). These results suggest the presence of the collapsed phase as the product of aerial surface oxidation."

4. What is the reason why the novel layered compound $\text{Sr}_2\text{MnO}_2\text{S}_2$ containing $(\text{S}_2)^{2-}$ dimers and its intergrowth structure could not be synthesized in their single phases? For example, is it because the structural stability of $\text{Sr}_2\text{MnO}_2\text{S}_2$, intergrowth structure compounds, and precursor $\text{Sr}_2\text{MnO}_2\text{Li}_{1.9}\text{S}_2$ are energetically competing? It may be considered that by calculating the total energy and phonon dispersion of these compounds, it would be possible to evaluate the thermodynamic stability of these compounds and clarify this reason.

Response:

This is a difficult question to answer before carrying out comprehensive mechanistic studies of its intercalation chemistry using both theoretical and experimental approaches. At the moment, we suspect that the removal of Cu/Li intercalants was limited rather for kinetic reasons than thermodynamic stabilities for the following reasons: (1) large voltage hysteresis during electrochemical Li deintercalation from $\text{Sr}_2\text{MnO}_2\text{Li}_2\text{S}_2$ (Fig. 4d) and (2) spontaneous formation of the collapsed phase from the parent-type $\text{Sr}_2\text{MnO}_2\text{Li}_x\text{S}_2$ phases upon air exposure (Fig. C). As discussed in our responses above, such voltage hysteresis was commonly observed during oxidation of anionic species (See e.g. Li, B., Sougrati et al. *Nat. Chem.* **13**, 1070–1080 (2021)) and this kinetic barrier was sometimes associated with sluggish structural distortions. Considering its large structural transformation, it is possible that the formation of the collapsed phase involves significant kinetic barriers. It might be possible that the collapsed phase is not thermodynamically favourable compared to its chemical variants $\text{Sr}_2\text{MnO}_2\text{Li}_x\text{S}_2$ (parent type) or intergrowth phases, but in this case the parent-type $\text{Sr}_2\text{MnO}_2\text{Li}_x\text{S}_2$ would not turn into the collapse phase unless highly oxidizing conditions were applied. However, what we observed was the spontaneous oxidation of the parent-type $\text{Sr}_2\text{MnO}_2\text{Li}_x\text{S}_2$ into the collapsed phase, even under very mild oxidizing conditions like air exposure (See Fig. C in our response above). Complete conversion into the collapsed phase was blocked, possibly due to kinetic reasons, and required the reaction with disulfiram at elevated temperature (80 °C).

To fully understand what obstructed complete Cu/Li removals, we must therefore investigate how oxidation takes place and how the structures evolve during the formation of collapsed phase, employing various in-situ/ex-situ characterizations. We believe that such a comprehensive study should be published in separate full paper. It is also a good idea to compute total energy and phonon dispersion of the relevant compounds. However, this would equally require systematic and comprehensive investigation considering e.g. various Li content x in the parent-type $\text{Sr}_2\text{MnO}_2\text{Li}_x\text{S}_2$ ($1 < x < 2$) and intergrowth-type $[\text{Sr}_2\text{MnO}_2\text{S}_2][\text{Sr}_2\text{MnO}_2\text{Li}_x\text{S}_2]$. Li contents in these phases are directly linked to $\text{Mn}^{2+/3+}$ or S^{2-} oxidation states, and eventually affecting local structure and stability of the $\text{Sr}_2\text{MnO}_2\text{S}$ square lattice (See for example: H.-J. Koo et al. *Inorg. Chem.* 2019, 58, 21, 14769–14776). Given their extent and demanding nature, we prefer that these calculations will be done as a separate and mechanism-oriented full paper.

These comprehensive mechanistic study is ongoing in our group. To signal our intention, we added the following sentences in page 12:

“...In any case, such kinetic barriers also hint at why the chemical deintercalation at Step 6 (Fig. 1a) could not remove all Cu/Li cations from the final product. Further in-depth analyses of its reaction dynamics are currently ongoing, employing in- or ex-situ spectroscopic and diffraction techniques.”

A similar mention was added also to page 18-19:

“We are currently investigating how the competition between $Mn^{2+/3+}$ and S^{2-} redox evolves during Li (de)intercalation processes as well as the complex reaction dynamics both from thermodynamic and kinetic points of view.”

Minor point

I thought it might be good to move the sentence about the details of the Rietveld analysis described in the second paragraph on page 13 to Supporting Information. The stacking faults and random distribution of intergrowth structures of the "collapsed" phase discussed in this paragraph seemed to be already clarified from experiments of STEM and EDX elemental mapping. Therefore, the reason for repeating the Rietveld analysis seemed unclear. I thought that the authors had a great effort for the Rietveld analysis with the periodic structural model and carefully performed the analysis. However, in the end, the authors concluded that the structural model considered here could not reasonably reproduce the diffraction peak intensity. Therefore, it seemed to me that the claim from the Rietveld analysis here is ambiguous.

(END of comments)

Response:

We agree with the referee on the point that our attempts to take Cu/Li into account could not provide further information beyond STEM and EDX due to unconvincing Rietveld fit to the XRD pattern. Accordingly, we made the discussion shorter and more concise (See below). Nevertheless, we believe that the attempt must be mentioned in the main text so that readers will be able to understand why we could not provide a refined structure model with Cu and Li cations reproducing the experimental diffraction patterns. We thus did not removed the entire discussion from the main text.

We simplified the former half of the corresponding discussion in page 14 as follows:

“The presence of both parent phase and intergrowth layers alongside the collapsed layers encouraged us to attempt Rietveld refinements explicitly taking the presence of these residual Cu and Li cations into account. A custom-built code generated ~20000 unique 1500-layer supercell models containing different combinations of those three structures (See section 3.4 in the SI for details).”

Also, we added the sentences to relate the discussion to following NMR results:

“Bulk probes like XRD have an apparent limit to analyze stacking faults in such spatially heterogeneous samples, and they must be complemented by local probes.”

Similar point was emphasized in the conclusion (page 19):

“We modelled the presence of these residual Cu^+ and Li^+ cations using stacking faults in which monoclinic $Sr_2MnO_2Ch_2$ regions intergrow with parent-type $Sr_2MnO_2Cu_{1.5}S_2$ slabs, but our Rietveld analysis could not explain their compositional and structural heterogeneity. On the other hand, our local probe analyses using HAADF-STEM imaging and 7Li NMR revealed that substantial regions consisted of periodic intergrowths of $Sr_2MnO_2Ch_2$ and $Sr_2MnO_2(Cu,Li)_2S_2$ -type layers.”

Reviewer #2 (Remarks to the Author):

This is a high-quality work about topochemical reaction of layered oxychalcogenides. The author reported multi-step Cu-deintercalation of $\text{Sr}_2\text{MnO}_2\text{Cu}_{1.5}\text{Ch}_2$ yielding the collapsed phase with Ch_2 dimers. The (almost) full deintercalation of Cu from $\text{Sr}_2\text{MnO}_2\text{Cu}_{1.5}\text{Ch}_2$ phase is the first case, but a similar concept was already reported by the authors in the other layered oxychalcogenides (e.g. ref 14; $\text{La}_2\text{O}_2\text{Cu}_2\text{S}_2$). The impressive point in this work is the controlled chemical reactions: They used an organic reagent, disulfiram, which seems to work as a chemoselective agent toward Cu. Their characterization of the collapsed phase is well done in that they used several methods to determine the structural and chemical states. Especially, they carried out advanced Rietveld analysis to fit the XRD pattern with defects or stacking faults.

I consider that this work is important for the fundamental interests of solid-state chemistry fields, and is worth publishing in *Nat. Commun.*

Additional comment:

I feel that the five-step-reaction is one of the most important part of this work. But there are no figures to see how XRD patterns changes in each reaction (There are some XRD patterns in supporting information but they are not whole reactions). I recommend that the authors put the five XRD patterns in the main figure.

Response:

We appreciate the encouraging feedback and the useful suggestion from the referee. Following the referee's advice, we combined the XRD patterns at each synthetic step (Originally Fig. S1 and S3) with Fig. 1a. Accordingly, the original Fig 1b-d were separated as the independent Fig. 2.

Reviewer #3 (Remarks to the Author):

Sasaki and coworkers report on the topochemical manipulation of a layered mixed-metal oxychalcogenide compounds. Sequential reaction steps involving copper extraction and lithium intercalation/deintercalation result in a series of topochemically related intermediates, culminating in a “collapsed” structure with new sulfur/selenium dimers. This result is exciting in that it shows the rigorous extraction of intermediate layers can be carried out while still retaining key structural features (MnO layers) of the parent compound. The researchers are thorough in their treatment of this system, including detailed X-ray, neutron, and electron diffraction studies. The electron diffraction studies are especially illuminating as to the concurrent complexity and beauty of the final products.

This research is of general interest but will be especially appealing to solid state chemists, researchers interested in topochemistry, and those focused on complex redox processes in energy storage materials.

The research is appropriate for Nature Communications and can be published without revision.

Response:

We are delighted to hear the warm and encouraging comments from the referee. We revised arrangements of the figures (Fig 1 and 2) and added some discussions about the synthetic routes and the reaction mechanism, following the suggestion from other referees.

REVIEWERS' COMMENTS

Reviewer #1 (Remarks to the Author):

Referee report for the revised manuscript NCOMMS-22-52782-T

After reviewing the materials sent to me, I found that the manuscript has been improved well. It effectively highlights advantages and challenges associated with the development of new inorganic materials that contain S₂(or Se₂) dimers through anion-redox topochemical reactions, which is the primary focus of this study. Overall, the revised manuscript appears to be of good quality.

Although the content may be technical in nature, the revised manuscript provides valuable insights into the potential of the methods introduced in this study. It discussed how these methods could be extended in the future to synthesize materials from a new perspective, including inorganic compounds containing molecular anions. This information adds to the significance of the study and its prospects for further research.

Based on the revised manuscript, I think that it meets the high standards required for publication in Nature Communications. Besides, I would recommend a final check to ensure that the text is as error-free and consistent as possible.

NCOMMS-22-52782A

**Anion Redox as a Means to Derive Layered Manganese
Oxychalcogenides with Exotic Intergrowth Structures**

S. Sasaki *et al.*

Response to Reviews.

Reviewer #1 (Remarks to the Author):

Reviewer Comments:

“After reviewing the materials sent to me, I found that the manuscript has been improved well. It effectively highlights advantages and challenges associated with the development of new inorganic materials that contain S₂(or Se₂) dimers through anion-redox topochemical reactions, which is the primary focus of this study. Overall, the revised manuscript appears to be of good quality.

Although the content may be technical in nature, the revised manuscript provides valuable insights into the potential of the methods introduced in this study. It discussed how these methods could be extended in the future to synthesize materials from a new perspective, including inorganic compounds containing molecular anions. This information adds to the significance of the study and its prospects for further research.

Based on the revised manuscript, I think that it meets the high standards required for publication in Nature Communications. Besides, I would recommend a final check to ensure that the text is as error-free and consistent as possible.”

Author Response:

We are grateful for the positive comments of the reviewer

We have indeed checked through the manuscript for grammar, ease of understanding and consistency and have made several minor changes (also taking into account the editorial advice).